# On the geometry and topology of representations: The manifolds of modular addition

**Gabriela Moisescu-Pareja**[*]
McGill University, Mila

**Gavin McCracken**[*]
McGill University, Mila

**Harley Wiltzer**
McGill University, Mila

**Vincent Létourneau**
Université de Montréal, Mila

**Colin Daniels**
Independent

**Doina Precup**
McGill University, Mila, Google DeepMind

**Jonathan Love**
Leiden University

## Abstract

The Clock and Pizza interpretations, associated with architectures differing in either uniform or learnable attention, were introduced to argue that different architectural designs can yield distinct circuits for modular addition. In this work, we show that this is not the case, and that both uniform attention and trainable attention architectures implement the same algorithm via topologically and geometrically equivalent representations. Our methodology goes beyond the interpretation of individual neurons and weights. Instead, we identify all of the neurons corresponding to each learned representation and then study the collective group of neurons as one entity. This method reveals that each learned representation is a manifold that we can study utilizing tools from topology. Based on this insight, we can statistically analyze the learned representations across hundreds of circuits to demonstrate the similarity between learned modular addition circuits that arise naturally from common deep learning paradigms.

## 1 Introduction

As deep neural networks (DNNs) scale and begin to be deployed in increasingly high-stakes settings, it will be imperative to develop a concrete understanding of how these models perform computations and ultimately make decisions. Towards this end, research in mechanistic interpretability has focused on identifying sub-structures of these models—referred to as *circuits*—and understanding the function and formation of these circuits on the subtasks that they are responsible for. In order to extract a generalizable understanding of circuits, researchers have formulated a key hypothesis **universality** (Li et al., 2015; Olah et al., 2020), which suggests that similar networks trained on similar data will form similar circuits. On the other hand, the **manifold** hypothesis (Bengio et al., 2013; Goodfellow et al., 2016), suggests that representation learning consists of finding a lower-dimensional manifold for the data. If these hypotheses were to be false, the task of interpreting large scale models becomes dire—there would be little hope of identifying common patterns across initializations, architectures, and datasets.

Yet, recent work claimed totally disparate and disjoint circuits were learned by DNNs trained on **the exact same data** using modular addition $(a + b) \mod n = c$ (Zhong et al., 2023). Their results seemingly provide a counter-example to the universality hypothesis, thus suggesting the pursuit of learning interpretable principles that generalize across tasks may be doomed. In fact, this suggests that the identification of simple circuits in larger neural networks could be combinatorially difficult—if different small-scale models trained on one task can learn totally different circuits with no commonality, then large language models (LLMs) may learn many disjoint circuits for a task within their weights simultaneously.

---

[*]Equal contribution. {gabriela.moisescu-pareja, gavin.mccracken}@mail.mcgill.ca

Our primary contribution to interpreting modular addition, *i.e.* the dataset $(a + b) \bmod n = c$ is the resolution of the *seemingly* disparate circuits found by Zhong et al. (2023).

1. We show that, under certain conditions on the structure of learned embeddings, all networks studied by Zhong et al. (2023) and McCracken et al. (2025) learn preactivations having equivalent geometry and topology, which can be expressed in closed form;

2. These closed-form equations allow us to rigorously claim that all architectures we study universally make use of the same class of manifolds;

3. We introduce new tools, leveraging topological data analysis, to empirically validate these aforementioned conditions, providing extensive empirical evidence of shared representation geometry across networks.

Altogether, our results restore the possibility that the universality hypothesis is true, as Zhong et al. (2023)'s architectures are no longer shown to be a counter-example.

## 2 RELATED WORK

Deep learning (DL) research increasingly turns to mathematical tasks as controlled settings for investigating learning phenomena and studying the fundamentals of DL (Ghosh, 2025). Such tasks provide opportunities to (i) derive exact functional forms that yield theoretical insights beyond empirical studies (McCracken, 2021), (ii) analyze how agents discover novel algorithms such as matrix multiplication or sorting (Fawzi et al., 2022; Mankowitz et al., 2023), and (iii) develop methods to better interpret learned policies (Raghu et al., 2018). From toy models that capture superposition (Elhage et al., 2022) to formal analyses of in-context learning (Lu et al., 2024; 2025), training on math tasks has become a productive way to study DL fundamentals. Within this agenda, mechanistic interpretability has produced especially influential results. By reverse-engineering networks trained on group-theoretic problems—such as modular addition (Nanda et al., 2023; Chughtai et al., 2023; Gromov, 2023; Morwani et al., 2024; McCracken et al., 2025; Yip et al., 2024; He et al., 2024; Tao et al., 2025; Doshi et al., 2023), permutations (Stander et al., 2024), and dihedral multiplication McCracken et al. — researchers have uncovered mechanisms that speak to core hypotheses about representations (Huh et al., 2024), universality (Olah et al., 2020; Li et al., 2015), and algorithmic structure in DL (Eberle et al., 2025).

As the datasets of group multiplication aren't linearly separable and modular addition (Cyclic group multiplication) is very well studied, it has become a standard testbed for toy interpretability research. It is ideal for asking: "*What exactly do neural networks learn, and how is that computation represented internally?*". Two influential works stand out. First, Nanda et al. (2023) reverse-engineered transformers trained on modular addition and described their internal computations to illuminate the *grokking* phenomenon (Power et al., 2022), giving progress measures to predict it. Building on this, Chughtai et al. (2023) claimed that the algorithm generalized to all group multiplications. Second, Zhong et al. (2023) modified the transformer from Nanda et al. (2023) by interpolating between uniform and learnable attention. They claimed their networks learned two distinct circuits, either a *Pizza* or a *Clock* (described by Nanda et al. (2023)), proposing metrics to separate the two disjoint circuits.

Recently however, McCracken et al. (2025) showed via abstraction, that across both MLPs and transformers, models converge to one unifying divide-and-conquer algorithm that approximates the *Chinese Remainder Theorem* and matches its logarithmic feature efficiency. They found first-layer neurons are best fit by degree-1 trigonometric polynomials, with later layers requiring degree-2, in contrast to the interpretation of Nanda et al. (2023) which modeled neurons as degree-2 trigonometric polynomials in all layers.

While many works have focused on reverse-engineering specific algorithms in modular addition, relatively little has been done to systematically compare neural representations themselves. Tools from other domains–such as distributional hypothesis testing and topological data analysis (TDA)—offer complementary ways to characterize representations and may enrich mechanistic interpretability. For instance, distributional methods such as maximum mean discrepancy (MMD) Gretton et al. (2012) are rarely used in mechanistic interpretability, though widely applied elsewhere to quantitatively measure similarity, align domains Ghifary et al. (2014); Zhao et al. (2019), and test fairness

Deka & Sutherland (2023); Kong et al. (2025). Topological data analysis (TDA) offers a complementary view: Shahidullah (2022) used persistent homology to track how network layers preserve or distort input topology, and Ballester et al. (2024) surveyed TDA tools such as persistent homology and Mapper for analyzing architectures, decision boundaries, representations, and training dynamics.

## 3 SETUP AND BACKGROUND

The learning task we are interested in is the operation of the cyclic group, modular addition $(a, b) \mapsto a + b \mod n$ for $a, b \in \mathbb{Z}_n$. In our experiments we have used $n = 59$. We consider various neural network architectures but we will mostly refer to them by the name that was given to an interpretation of their neuronal operations. All architectures begin by embedding the inputs $a, b$ to vectors $\mathbf{E}_a, \mathbf{E}_b \in \mathbb{R}^{128}$ using a shared (learnable) embedding matrix. The architectures differ in how the embeddings are then processed: **MLP-Add** immediately passes $\mathbf{E}_a + \mathbf{E}_b$ through an MLP, **MLP-Concat** immediately passes the concatenation $\mathbf{E}_a \oplus \mathbf{E}_b \in \mathbb{R}^{256}$ through an MLP. Learnable attention (claimed to learn **Clocks**) and uniform attention (claimed to learn **Pizzas**) transformers, introduced by Zhong et al. (2023), pass $\mathbf{E}_a, \mathbf{E}_b$ through a self-attention layer before the MLP. Specifically: uniform attention is a constant attention matrix, and trainable attention (Nanda et al., 2023) uses the standard learnable softmax attention. We refer to trainable attention architectures as **Attention 1.0** and constant attention architectures as **Attention 0.0** since Zhong et al. (2023) used a parameter to switch between them.

### 3.1 PREVIOUS INTERPRETATIONS OF MODULAR ADDITION IN NEURAL NETWORKS

Prior works have all reported disjoint circuits of different "frequencies" $f$ are learned. We adopt the notation of Zhong et al. (2023), each frequency $f$ is associated with a circuit, and an abstraction can be given for the values in the embedding vectors $\mathbf{E}_a$ and $\mathbf{E}_b$ associated with that frequency,

$$\mathbf{E}_a = [\cos(2\pi f a/n), \sin(2\pi f a/n)], \quad \mathbf{E}_b = [\cos(2\pi f b/n), \sin(2\pi f b/n)]. \quad (1)$$

What distinguishes them is how the embeddings are *transformed* post-attention. Treating the attention as a blackbox and looking at its output $\mathbf{E}_{ab}$, Zhong et al. (2023) make the following two claims. **Clock** circuits compute the *angle sum* (Figure 1),

$$\mathbf{E}_{ab} = [\cos(2\pi f(a + b)/n), \sin(2\pi f(a + b)/n)], \quad (2)$$

encoding the modular sum on the unit circle, which needs second-order interactions (e.g., multiplying embedding components via sigmoidal attention). In **Pizza** circuits, $\mathbf{E}_{ab}$ adds the embeddings directly as $\mathbf{E}_a + \mathbf{E}_b$,

$$\mathbf{E}_{ab} = [\cos(2\pi f a/n) + \cos(2\pi f b/n), \sin(2\pi f a/n) + \sin(2\pi f b/n)], \quad (3)$$

yielding *vector addition* on the circle (Figure 1), which is entirely linear in the embeddings. Zhong et al. (2023) gave metrics distance irrelevance and gradient symmetricity to distinguish networks having learned the clock vs. pizza circuit (see Appendix E).

**Definition 3.1** (Simple Neurons). For $(a, b) \in \mathbf{Z}_p \times \mathbf{Z}_p$, a *simple neuron* is a neuron that has pre-activation

$$N(a, b) = \cos(2\pi f a/n + \phi^L) + \cos(2\pi f b/n + \phi^R), \quad (4)$$

where $f \in \mathbf{Z}_p$ is a frequency and $\phi^L, \phi^R \in [0, 2\pi)$ are phase shifts.

This form of the neurons as being a linear superposition of a sinusoid in $a$ and a sinusoid in $b$ is corroborated in Gromov (2023); Morwani et al. (2024); Doshi et al. (2023); McCracken et al. (2025); Li et al. (2025).

**Empirical Fact 3.2** (McCracken et al. (2025)). For **Attention 0**, **Attention 1**, and **MLPs** (with learnable or one-hot encoded embeddings) architectures, first layer neurons are well approximated by *simple neurons*. Later layers can encode combinations of degree-1 and 2 sinusoids.

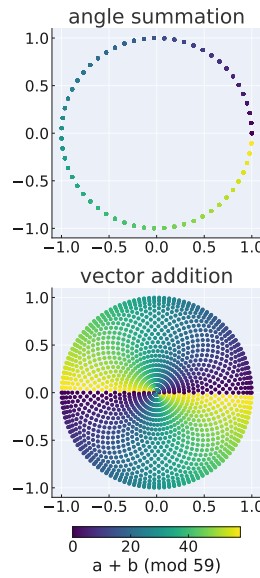

angle summation

vector addition

$a + b \pmod{59}$

Figure 1: Clock and Pizza's analytical forms visualized, with frequency assumed to be $f = 1$ for simplicity. Each point corresponds to a pair $(a, b)$ after being transformed by the corresponding analytical form and is colored by its sum $(a + b)$ mod 59.

## 3.2 REPRESENTATION SIMILARITY MEASURES

We use two representation similarity measures: centered kernel alignment (CKA) (Kornblith et al., 2019) and the representational similarity matrix (RSM) (Kriegeskorte et al., 2008).

## 3.3 TOPOLOGICAL DATA ANALYSIS

We use **Betti numbers** from algebraic topology to distinguish the structure of different stages of circuits across layers. The $k$-th Betti number $\beta_k$ counts $k$ dimensional holes: $\beta_0$ counts connected components, $\beta_1$ counts loops, $\beta_2$ counts voids enclosed by surfaces. For reference, a disc has Betti numbers $(\beta_0, \beta_1, \beta_2) = (1, 0, 0)$, a circle has $(1, 1, 0)$, and a 2-torus has $(1, 2, 1)$.

## 4 CANONICAL MANIFOLDS

We will now focus on networks with a single learnable embedding matrix, matching the setups of Nanda et al. (2023); Zhong et al. (2023); McCracken et al. (2025). Our analysis will center on the *representation manifolds* in a frequency cluster $f$ coming from the preactivations $h_{\ell,f}^{\text{pre}}(a, b)$ at layer $\ell$ and the logits $l_f(a, b)$. The corresponding representation manifolds are, explicitly,

$$\mathcal{M}_{\ell,f}^{\text{pre}} := \left\{ h_{\ell,f}^{\text{pre}}(a, b) : (a, b) \in \mathbb{Z}_n^2 \right\} \subset \mathbb{R}^{d_{\ell,f}}; \quad \text{and} \quad \mathcal{M}_f^{\text{logit}} := \left\{ l_f(a, b) : (a, b) \in \mathbb{Z}_n^2 \right\} \subset \mathbb{R}^n,$$

where $d_{\ell,f}$ is the number of neuron in the frequency cluster $f$ at layer $\ell$. Our thesis is that under the simple neuron model of equation 4 introduced by McCracken et al. (2025) and a simple application of symmetry corresponding to the interchangeability of $a, b$ in $a + b \mod n$, the exact structure of the $\mathcal{M}_{1,f}^{\text{pre}}$ manifolds and how they are mapped from inputs $a, b$ can be revealed. Particularly, we will show that under this model, $\mathcal{M}_{1,f}^{\text{pre}}$ always encodes the torus $\mathbb{T}^2$ or vector addition disk of Figure 1—that is, the pizza.

The remainder of this section will proceed by formalizing this result in §4.1.

## 4.1 SIMPLE NEURON PHASE DISTRIBUTION DICTATES REPRESENTATION MANIFOLD

Under the simple neuron model, for any frequency cluster $f$, the only degrees of freedom in the resulting preactivations lie in the maps $(a, b) \mapsto (\phi^L, \phi^R)$ for $a, b \in \mathbb{Z}_n$ learned by neural networks. Given that modular addition is commutative, one might expect to see a form of symmetry with respect to $\phi^L, \phi^R$. Particularly, one might expect that $\phi^L \equiv \phi^R$ for all $a, b$ (since swapping the inputs should have no effect on the output), or at the very least that the random variables $\phi^L, \phi^R$ are identically distributed for $A, B \sim \text{Uniform}(\mathbb{Z}_n)$. It turns out, as we show in the following theorem (whose proof is given in Appendix B), that the resulting manifold $\mathcal{M}_{1,f}^{\text{pre}}$ takes an easily characterizable form *almost surely* in this event. We devote §6 to validating that the phase maps indeed satisfy these properties in practice—allowing us to easily analyze the geometry of representations across thousands of trained neural networks.

Before stating the theorem, let us introduce some notation that will be useful. Under the simple neuron model, a neuron indexed $i$ belonging to a neuron cluster with frequency $f$ maps $(a, b) \in \mathbf{Z}_p^2$ to $\cos(\theta_a + \Phi_i^L) + \cos(\theta_b + \Phi_i^R)$, where $\theta_a = 2\pi f a / p$. The notation $\Phi_i$ is meant to evoke that we model these phases as random variables; these are random due to random initialization and random gradient updates. The joint distribution of $(\Phi_i^L, \Phi_i^R)$ is denoted $\mu_i^{a,b} \in \Delta([0, 2\pi]^2)$.

**Theorem 4.1.** *Let $f \in \mathbf{Z}_p$ for $p \geq 3$, and consider the frequency cluster at layer 1. Let $m$ denote the number of neurons in this cluster, and assume $m \geq 2$. Define the matrix $X \in \mathbb{R}^{p^2 \times m}$ according to $X_{(a,b),i} = \cos(\theta_a + \phi_i^L) + \cos(\theta_b + \phi_i^R)$, denoting the simple neuron preactivations. Assume $\phi_i^L, \Phi_i^{L,b}$ are identically distributed for each neuron $i \in \{1, \ldots, m\}$ in this cluster, and that the support of $\mu_i^{a,b}$ has positive (Lebesgue) measure. Then the following hold almost surely:*

1. *(Perfect phase correlation) If $\Phi_i^{L,a}$ and $\Phi_i^{R,b}$ are perfectly correlated, in the sense that $\Phi_i^{L,b} \equiv \Phi_i^{R,b}$, then $X$ has a rank-2 factorization $X = V^{\text{disc}}W$ with $V^{\text{disc}} \in \mathbb{R}^{p^2 \times 2}$ satisfying*

$$V_{(a,b)}^{\text{disc}} = (\cos\theta_a + \cos\theta_b, \sin\theta_a + \sin\theta_b)^\top. \tag{5}$$

2. (Phase independence) *Otherwise, $X$ has a rank-4 factorization $X = V^{\text{torus}}W$ with $V^{\text{torus}} \in \mathbb{R}^{p^2 \times 4}$ given by*

$$V^{\text{torus}}_{(a,b)} = (\cos\theta_a, \sin\theta_a, \cos\theta_b, \sin\theta_b)^\top. \tag{6}$$

Geometrically, the disc can be viewed as a projection of the torus: $(x_1, x_2, x_3, x_4) \mapsto (x_1 + x_3, x_2 + x_4)$. Thus, the torus structure generalizes the vector-addition disc.

Having established this theorem, it is worth stepping back to contextualize its consequences. As shown by McCracken et al. (2025), first layer preactivations are dominantly simple neurons. Theorem 4.1 shows that, under the symmetry properties of $\Phi_i^{L,a}$ and $\Phi_I^{R,b}$ posited above, the preactivations have simple, low-dimensional structures: in the case of perfect phase correlation, the representation manifold can be compressed to $V^{\text{disc}}$, which is precisely the vector addition disc of Figure 1. In the case of phase independence, the representation manifold can be compressed to $V^{\text{torus}}$, which exactly encodes the torus $\mathbb{T}^2$.

**Remark 4.2.** It is noteworthy that the Clock representation from Zhong et al. (2023) *cannot* occur under the hypotheses of Theorem 4.1. The remainder of the paper demonstrates that these hypotheses are satisfied empirically with overwhelming probability. Thus, while the Clock circuit of Zhong et al. (2023) is theoretically plausible, it does not occur naturally in practice. On the other hand, the possibility of the *torus* representation has not previously been identified in the literature.

A notable consequence of this result is that the geometry and topology of representation manifolds can be characterized by simply investigating the distributions $\mu_i^{a,b}$ of the learned phases. As we describe in §5, this can be done quantitatively, allowing us to derive statistical likelihoods of neural circuits arising over thousands of initializations across architectures.

## 4.2 QUALITATIVE ANALYSIS OF INTERMEDIATE REPRESENTATIONS

This section presents the experimental observations that support the predictions of section 4.1. Given that learned embeddings are expected to be points on a circle, we consider two MLP-models as simple baselines capturing the closed form of the above manifolds: MLP-Add should add two points on a circle, giving the vector addition disc (Figure 1) and MLP-Concat should concatenate two points on a circle, giving the torus $\mathbb{T}^2$.

In all networks, we cluster neurons together and study the entire cluster at once. This is done by constructing an $n \times n$ matrix, with the value in entry $(a, b)$ corresponding to the preactivation value on datum $(a, b)$. A 2D Discrete Fourier Transform (DFT) of the matrix gives the key frequency $f$ for the neuron. The cluster of preactivations of all neurons with key frequency $f$ is the $n^2 \times |$cluster $f|$ matrix, made by flattening each neurons preactivation matrix and stacking the resulting vector for every neuron with the same key frequency. We call this matrix the neuron-cluster of preactivations matrix.

**Principal component analysis (PCA).** To probe the representational geometry of the first-layer neurons, we perform PCA on the neuron-cluster matrix of pre-activations. This layer is where differences between Clock and Pizza architectures are expected to emerge, since their outputs diverge only after attention.

Figure 2 shows the PCA embeddings for MLP-Add, MLP-Concat, Pizza, and Clock models. Each input pair $(a, b)$ has been remapped following (McCracken et al., 2025) and coloured by $a + b \mod n$, so that points close with respect to the modular addition task are visually grouped.

The results align strikingly with the predictions of Theorem 4.1. For MLP-Add, Pizza, and Clock, the first two principal components explain more than 99% of the variance, yielding a 2D disc-like structure (matching the vector addition geometry of Figure 1). For MLP-Concat, the first four principal components each explain about 25% of the variance, producing a toroidal structure.

Surprisingly, the Clock network—never previously described as having a "pizza disc" exhibits the same 2D disc structure as Pizza and MLP-Add.

**Distribution of post-ReLU activations.** We next examine the strength of activations across neurons within a cluster. For each neuron, we construct an $n \times n$ heatmap of its post-ReLU activation

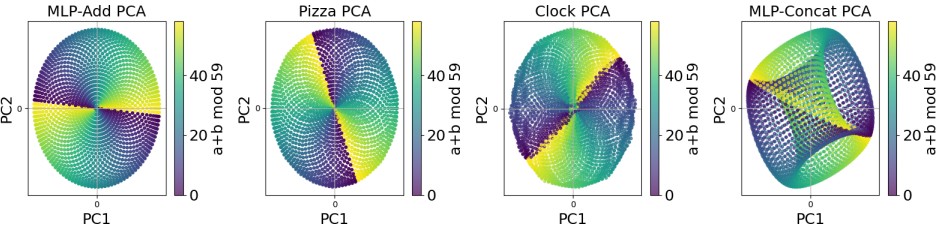

Figure 2: PCA of neuron pre-activations for a single frequency cluster across architectures: MLP-Add ($f = 27$), Pizza ($f = 17$), Clock ($f = 21$), MLP-Concat ($f = 22$). Each point is an input $(a, b)$, colored by $(d \cdot a + d \cdot b) \bmod 59$ corresponding to the network's output (see Section 3.1). Pizza and Clock are nearly identical to each other and to MLP-Add, but differ strongly from MLP-Concat. Note that the trained pizza and clock models being PCA'd are downloaded directly from Zhong et al. (2023)'s Github: model_p99zdpze5l.pt and model_l8k1hzciux.pt.

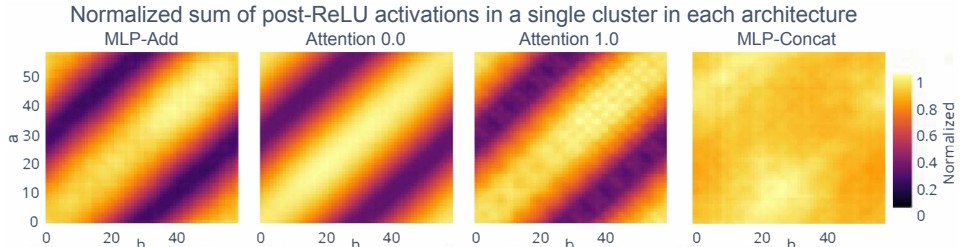

Figure 3: MLP-Add(f=27), Pizza(f=17), Clock(f=21), MLP-Concat(f=22). Normalized sum of post-activations in clusters in each architecture over all $(a, b)$. Clusters in MLP vector add, Attention 0.0 and 1.0 activate strongest on $(a, b)$ with $a$ close to $b$: the activation strength decreases with distance from $a = b$. Clusters in MLP-Concat activate almost equivariantly.

values over all input pairs $(a, b)$, after applying the remapping procedure of (McCracken et al., 2025). Each heatmap thus reflects the activation profile of a single neuron across the input space.

To summarize activation patterns at the cluster level, we sum the heatmaps of all neurons in the cluster. Although the network never performs this sum directly, it provides a compact visualization of where activation strength is concentrated across the input domain.

Figure 3 shows that in MLP-Add, Clock, and Pizza, activations are sharply concentrated along the $a = b$ diagonal. This indicates that cosine responses peak when $a = b$, consistent with neurons having equal phases. Moreover, the smooth falloff of activations away from the diagonal implies that representations also encode the distance $|a - b|$. Because downstream weights from neurons to logits are fixed across inputs, this structure implies that logits systematically depend both on $(a, b)$ and on the separation $a - b$.

Surprisingly, this diagonal dependence—previously claimed by Zhong et al. (2023) to be the defining characteristic of Pizza—appears not only in Pizza networks but also in MLP-Add and Clock. We return to this point in the experiments section.

**Linking PCA geometry and activation distributions.** The PCA results and activation heatmaps reflect the same underlying phenomenon: principal components encode activation strength. In Figure 2, MLP-Add, Pizza, and Clock show negligible activation near the origin $(0, 0)$ of the PCA plane, since neither principal component contributes strongly there. By contrast, for MLP-Concat the 4D embedding contains no datapoints near $(0, 0, 0, 0)$, indicating that the cluster activates roughly equally across all inputs. This observation supports the view that the torus-to-circle map is the natural map for MLP-Concat (Appendix H).

More generally, these findings suggest that activation strength determines how inputs are arranged in PCA space. For MLP-Add, Pizza, and Clock, the dependence on $a - b$ produces an annular geometry at the logits rather than a perfect circle (Figure 4). For MLP-Concat, the absence of $a - b$ dependence yields a representation closer to a true circle, consistent with the circular structure

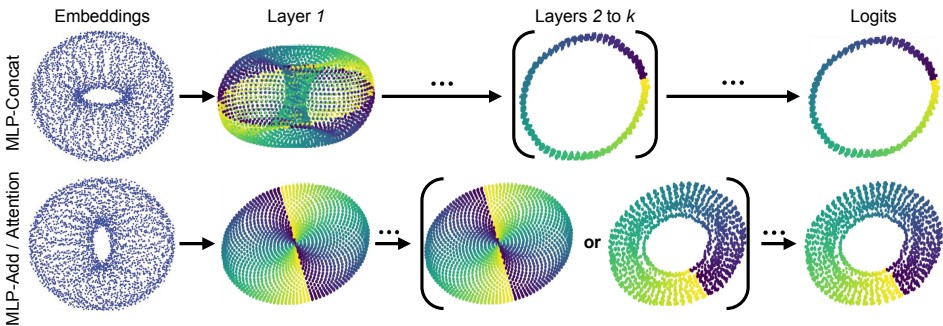

Figure 4: Different factorizations of the torus-to-circle map. We find first-layer intermediate representations to be either a torus or a disc (resembling vector addition on the circle). Later layers can construct a circle, and the logits approximate a circle. See Appendix H regarding the torus-to-circle factorization.

given in the torus-to-circle map (Appendix H) and illustrated in Figure 1. Figure 4 confirms this distinction: MLP-Concat exhibits a torus with a nearly uniform circular projection, while MLP-Add, Pizza, and Clock exhibit a vector addition disc (pizza) with an annular structure at the logits.

Finally, Figure 4 shows the possible manifolds we see in networks across layers.

## 5 METHODOLOGY

**Phase Alignment Distributions.** Given Theorem 4.1, we can classify representation manifolds from phase statistics. Thus, we propose yet another representation: the Phase Alignment Distribution (PAD). To a given architecture, a PAD is a distribution over $\mathbb{Z}_n \times \mathbb{Z}_n$. Samples of this distribution are drawn as follows:

1. Sample a random initialization (e.g., random seed) and train the network.

2. From the resulting trained network, sample a neuron uniformly.

3. Return the pair $(a, b) \in \mathbb{Z}_n \times \mathbb{Z}_n$ that achieves the largest activation in the resulting neuron.

A PAD illustrates, across independent training runs and neuron clusters, how often activations are maximized on the $a = b$ diagonal—that is, it depicts how often learned phases align. Even beyond inspecting the proximity of samples to this diagonal, we propose to compare the PADs of architectures according to a metric on the space of distributions over $\mathbb{Z}_n \times \mathbb{Z}_n$, giving an even more precise comparison. In the following section, we will provide estimates of the PADs for the aforementioned architectures, as well as PAD distances under the *maximum mean discrepancy* (Gretton et al., 2012, MMD)—a family of metrics with tractable unbiased sample estimators.

Following prior work on one-layer networks (Nanda et al., 2023; Zhong et al., 2023), and building on the empirical validation of the simple neuron model (Eq. 4) in McCracken et al. (2025), we restrict our PAD analysis to single-hidden-layer architectures. In practice, raw activations are often noisy, which makes direct sinusoid fitting unreliable. Instead, we assume the simple neuron model and extract the learned phase pair $(\phi^L, \phi^R)$ using one of two procedures: (i) by taking the point of maximal activation, or (ii) by computing the activation's center of mass. Both estimators yield qualitatively similar PADs; we report any differences between them in the Results section.

**Betti number distribution.** Now we turn to multi-layer networks. We estimate the distribution over Betti number vectors corresponding to the set of neurons within a cluster in a given layer (or logits) to distinguish the structure of the layers: that is $\mathcal{M}_{\ell,f}$ and $\mathcal{M}_{\text{logits}_f}$. This helps us identify when the underlying structure resembles a disc, torus, or circle. We compute these using **persistent homology** with the Ripser library (Bauer, 2021; de Silva et al., 2011; Tralie et al., 2018). For more details see Appendix A.3.

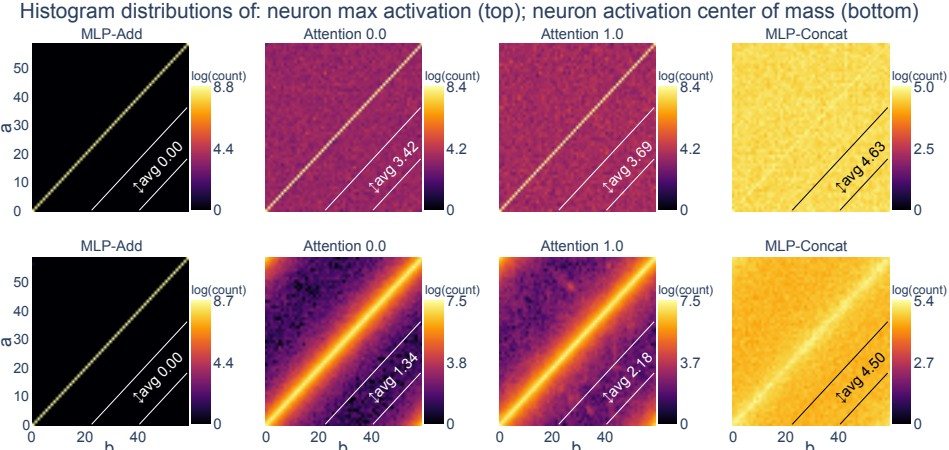

Figure 5: Log-density heatmaps for the distribution of neuron maximum activations (top) and activation center of mass (bottom) across 703 trained models. Attention 0.0 and 1.0 architectures exhibit modest off-diagonal spread compared to MLP-Add, but remain constrained by architectural bias toward diagonal alignment. The maximum mean discrepancy scores between Attention 0.0 and 1.0 are 0.0237 and 0.0181 in rows 1 and 2 respectively, indicating they are very similar distributions.

## 6 RESULTS

### 6.1 MLP-ADD, ATTENTION 0.0 AND 1.0 (PIZZA AND CLOCK) ARE *nearly* THE SAME

**PAD results.** We study 703 trained one-hidden-layer networks drawn from our four architectures: MLP-Add, Attention 0.0 (Pizza), Attention 1.0 (Clock) and MLP-Concat.

Figure 5 shows PAD plots across architectures. We note that MLP-Add, Attention 0.0, Attention 1.0 are very concentrated on the diagonal, while MLP-Concat is not. To further quantify this, we propose the torus distance, which is the discrete graph distance from a point $(a, b)$ on the torus to the $a = b$ line. Figure 7 quantifies this with a histogram of torus distances to the $a = b$ line. We see that Attention 0.0 and 1.0 are almost indistinguishable and both very similar to MLP-Add; moreover, our metric successfully discerns these models from MLP-Concat.

Table 4 shows the PAD distances under the MMD distance. All comparisons are statistically significant (p-values ≈ 0). We see that Attention 0.0 and 1.0 are extremely close to each other, MLP-Add lies moderately close to both, and MLP-Concat is strongly separated from all others. We also introduce and study the *torus distance* metric in Appendix D.

**Previous metrics.** In Appendix E (particularly E.2) we evaluate the metrics gradient symmetricity and distance irrelevance of Zhong et al. (2023).

**CKA and RMS results.** We report similarity using centered kernel alignment (CKA) and representational similarity matrix correlation (RSM). Linear CKA is bounded in [0,1], where 1 indicates identical representational geometry (up to orthogonal transformation and isotropic scaling) and 0 indicates no linear similarity. RSM values correspond to the correlation between pairwise similarity matrices and are bounded in [-1,1] Thus, values approaching 1 reflect strong alignment between learned representations and the reference manifold, while values near zero indicate weak or absent geometric similarity.

We report results for 1-layer networks compared to reference manifolds predicted by Theorem 4.1. Table 6.1 shows that first-layer representations strongly reflect the inductive bias of each architecture: MLP-Concat aligns almost entirely with the torus manifold, whereas MLP-Add aligns with the disc (vector-addition) manifold. Both attention models (Attention 0.0 and Attention 1.0) likewise exhibit representations that are predominantly disc-like. In contrast, Table 6.1 shows that the logits across models are overwhelmingly aligned with the circle (clock) manifold.

Table 1: Layer 1 representations: representation similarity measures compared to specific reference manifolds (mean $\pm$ std).

| Reference Manifold | Metric | MLP-Concat | MLP-Add | Attn_0.0 | Attn_1.0 |
|---|---|---|---|---|---|
| DISC (VECTOR ADD) | CKA | $0.707 \pm 0.012$ | $0.998 \pm 0.017$ | $0.988 \pm 0.041$ | $0.974 \pm 0.028$ |
| | RSM | $0.578 \pm 0.015$ | $0.998 \pm 0.018$ | $0.986 \pm 0.044$ | $0.972 \pm 0.034$ |
| TORUS | CKA | $0.994 \pm 0.011$ | $0.706 \pm 0.012$ | $0.699 \pm 0.029$ | $0.689 \pm 0.020$ |
| | RSM | $0.991 \pm 0.016$ | $0.579 \pm 0.010$ | $0.576 \pm 0.022$ | $0.581 \pm 0.014$ |
| CIRCLE (CLOCK) | CKA | $1.60\mathrm{e}{-11} \pm 2.12\mathrm{e}{-10}$ | $-3.84\mathrm{e}{-12} \pm 3.00\mathrm{e}{-10}$ | $-1.01\mathrm{e}{-10} \pm 7.60\mathrm{e}{-10}$ | $0.012 \pm 0.010$ |
| | RSM | $0.112 \pm 0.003$ | $0.109 \pm 0.002$ | $0.109 \pm 0.004$ | $0.126 \pm 0.014$ |

Table 2: Cluster contributions to logits: representation similarity measures compared to specific reference manifolds (mean $\pm$ std).

| Reference Manifold | Metric | MLP-Concat | MLP-Add | Attn_0.0 | Attn_1.0 |
|---|---|---|---|---|---|
| DISC (VECTOR ADD) | CKA | $4.61\mathrm{e}{-4} \pm 0.001$ | $0.037 \pm 0.094$ | $0.002 \pm 0.005$ | $0.002 \pm 0.004$ |
| | RSM | $0.110 \pm 0.006$ | $0.300 \pm 0.148$ | $0.258 \pm 0.047$ | $0.209 \pm 0.060$ |
| TORUS | CKA | $0.001 \pm 0.001$ | $0.026 \pm 0.066$ | $0.001 \pm 0.003$ | $0.002 \pm 0.003$ |
| | RSM | $0.115 \pm 0.003$ | $0.139 \pm 0.056$ | $0.115 \pm 0.007$ | $0.115 \pm 0.008$ |
| CIRCLE (CLOCK) | CKA | $0.986 \pm 0.025$ | $0.926 \pm 0.064$ | $0.940 \pm 0.071$ | $0.941 \pm 0.078$ |
| | RSM | $0.981 \pm 0.033$ | $0.881 \pm 0.097$ | $0.917 \pm 0.086$ | $0.925 \pm 0.091$ |

**Betti number results.** It's the case that the homology of what networks learn gives that MLP-Add, Attention 0.0 and Attention 1.0 architectures are all making topologically equivalent computations. While the MLP-Concat model appears to be different, it's in fact just more efficient, which results from the torus already having the holes necessary to accurately project the correct answer onto the logits after just one non-linearity (see Fig. 6). It's worth noting that while discs appear to be learned in the logits, in all the cases checked by hand the discs were caused by limitations of persistent homology, which struggles to find a hole of small radius .

## 7 DISCUSSION, LIMITATIONS, AND CONCLUSION

This work introduces a new lens under which circuits for modular addition may be compared. We argue that networks trained on modular addition tend to learn a common logit manifold, and that representations at intermediate layers are dictated by the alignment of learned phases for the two inputs in the first layer preactivations. Studying the distribution over phases, we identify that networks learn torus or vector-addition disc manifolds (known colloquially as *pizzas*) in the first layer, and proceed by iteratively applying rotations and linear projections to these manifolds before ultimately arriving at a logit annulus. We argue our work recovers the universality hypothesis by showing that the counter-example architectures of Zhong et al. (2023), and even our torus-learning MLP-Concat architecture all use the same torus to logits map. The transformers and MLP-Add architecture simply learn lower rank manifolds of the same class of torus manifolds. This follows rigorously and directly from our closed-form equations for the torus and pizza discs: the pizza discs are a linear projection of the torus. Thus, we posit "is the nature of universality that DNNs recover either a universal manifold, or linear projections of it in order to fit data?

Notably, this conclusion was reached by examining networks from a different perspective, which allowed us to mathematically predict the structure of representations, and also to test these predictions precisely at large scale using tools from topological data analysis. In particular, our work identifies that the joint distribution over simple neuron phases both determines the geometry of representations, and permits efficient quantitative evaluation.

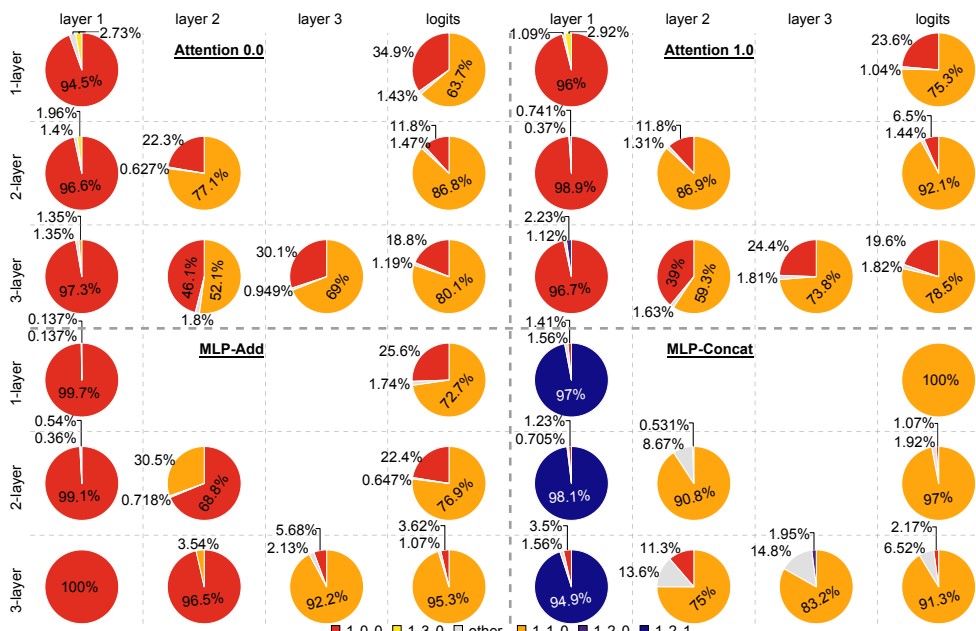

Figure 6: Betti number distributions across layers for 1-, 2-, and 3-layer models (100 seeds for each model). In layer 1, MLP-Add, Attention 0.0, and Attention 1.0 mostly yield disc-like representations, while MLP-Concat produces a torus. From the second layer onward, MLP-Add and both Attention variants converge to either a disc or a circle: the circle reflects the logits topology (correct answer), while the disc is an intermediate that can persist in later layers. MLP-Concat instead transitions directly to the circle. Across depth, Attention 0.0 and 1.0 are nearly identical with the latter having fewer transient discs.

A notable limitation of our work is that, in its present form, it treats only circuits in modular addition. To interpret large-scale neural networks, it will be necessary to derive strategies for characterizing circuits across domains. While our precise methodology is particular to the analysis of modular addition networks, we believe the underlying strategy of isolating the degrees of freedom in the geometry of hidden representations and estimating the statistics of their corresponding learned values can be a generalizable approach to characterize circuits in more complex domains.

Beyond this, our results also suggest that the universality property is likely to hold in modular addition. It also draws connections between universality and the manifold hypothesis, given the correspondence we see between the low-dimensional representations formed throughout the layers of the networks we observed. Ultimately, we believe the tools we introduced in this work can inform formal hypotheses about universality and manifold continuity that can be assessed quantitatively.

## REPRODUCIBILITY STATEMENT

We have provided our code in a supplementary.zip and experimental settings in Appendix A. We have also provided GPU optimized procedures for all computations we perform including the reproduction of Zhong et al. (2023)'s metrics to make repeating our tests more accessible.

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

## A    ADDITIONAL EXPERIMENTAL SETUP DETAILS

### A.1    TRAINING HYPERPARAMETERS.

All models are trained with the Adam optimizer Kingma & Ba (2014). Number of neurons per layer in all models is 1024. Batch size is 59. Train/test split: $90\%/10\%$.

**Attention 1.0**

- Learning rate: 0.00075
- L2 weight decay penalty: 0.000025

**Attention 0.0**

- Learning rate: 0.00025
- L2 weight decay penalty: 0.000001

**MLP-Add and MLP-Concat**

- Learning rate: 0.0005
- L2 weight decay penalty: 0.0001

### A.2    CONSTRUCTING REPRESENTATIONS

In all networks, we cluster neurons together and study the entire cluster at once McCracken et al. (2025). This is done by constructing an $n \times n$ matrix, with the value in entry $(a, b)$ corresponding to the preactivation value on datum $(a, b)$. A 2D Discrete Fourier Transform (DFT) of the matrix gives the key frequency $f$ for the neuron. The cluster of preactivations of all neurons with key frequency $f$ is the $n^2 \times |\text{cluster } f|$ matrix, made by flattening each neurons preactivation matrix and stacking the resulting vector for every neuron with the same key frequency.

### A.3    PERSISTENT HOMOLOGY

We compute these using **persistent homology**, applied to point clouds constructed from intermediate representations at different stages of the circuit, as well as the final logits. This yields a compact topological signature that captures how the geometry of these representations evolves across layers, helping us identify when the underlying structure resembles a disc, torus, or circle. We use the Ripser library for these computations Bauer (2021); de Silva et al. (2011); Tralie et al. (2018).

For our persistent homology computations, we set the $k$-nearest neighbour hyperparameter to 250. Our point cloud consists of $59^2 = 3481$ points.

### A.4 REMAPPING PROCEDURE

**Neuron remapping (McCracken et al., 2025).** For a simple neuron of frequency $f$, we define a canonical coordinate system via the mapping:

$$(a, b) \mapsto (a \cdot d, b \cdot d), \qquad \text{where } d := \left(\frac{f}{\gcd(f,n)}\right)^{-1} \mod \frac{n}{\gcd(f,n)}. \tag{7}$$

This inverse is the modular multiplicative inverse, i.e. for any $\mathbb{Z}_k$ let $x \in \mathbb{Z}_k$. Its inverse $x^{-1}$ exists if $\gcd(x, k) = 1$ and gives $x \cdot x^{-1} \equiv 1 \mod k$. This normalizes inputs relative to the neuron's periodicity and allows for qualitative and quantitative comparisons.

## B PROOF OF THEOREM 4.1

The proofs of both cases of Theorem 4.1 follow the same pattern: apply an angle sum formula to the entries of the pre-activation matrix, realize this matrix as a product of 2 low-rank matrices and use the assumption of uniformity of the phase variables to deduce full rank of the composition.

For integers $p \geq 3$ and $m \geq 2$, consider the $p^2 \times m$ data matrix of the pre-activations of the model network with simple neurons (seee equation 4 and 1).

$$X_{(a,b),i} = \cos(\theta_a + \Phi_i^{L,a}) + \cos(\theta_b + \Phi_i^{R,b}), \qquad \theta_t := \frac{2\pi t}{p}, \ (a,b) \in \{0, \dots, p-1\}^2.$$

and using the identity $\cos(x + y) = \cos x \cos y - \sin x \sin y$, we have

$$X_{(a,b),i} = \cos(\theta_a)\cos(\Phi_i^{L,a}) - \sin(\theta_a)\sin(\Phi_i^{L,a}) + \cos(\theta_b)\cos(\Phi_i^{R,b}) - \sin(\theta_b)\sin(\Phi_i^{R,b}) \tag{8}$$

Next, we show the details specific to each of the cases: disc and torus.

**Proof of Theorem 4.1 (Disc)**

*Proof.* By assumption, $\Phi_i^{L,a} = \Phi_i^{R,b} = \phi_i$ for all $i$. Then, equation 8 becomes

$$X_{(a,b),i} = (\cos\theta_a + \cos\theta_b)\cos\phi_i - (\sin\theta_a + \sin\theta_b)\sin\phi_i \tag{9}$$

Notice then that $X = VW$ for the matrices $V$ and $W$ defined by the following row and column vectors respectively

$$V_{(a,b),:} := [\cos(\theta_a) + \cos(\theta_b), \sin(\theta_a) + \sin(\theta_b)] \tag{10}$$

$$W_{:,i} := \begin{bmatrix} \cos(\phi_i) \\ -\sin(\phi_i) \end{bmatrix}. \tag{11}$$

Now we show they both have rank 2 and the kernel of $W$ intersect the image of $V$ trivially. The rank 2 of $V$ follows from the independence of $\cos$ and $\sin$ and the rank 2 of $W$ is true almost surely following the the independence of $\cos$ and $\sin$ and the hypothesis that $\Phi_i^{L,a}$ and $\Phi^{R,b}$ have uncountable support.

Suppose $\langle V_{(a,b),:}, W_{:,i}\rangle = 0$ for some $(a,b)$ and all $i$, that means $\cos(\theta_a + \phi_i) = -\cos(\theta_b + \phi_i)$ for all $i$. From the assumption the random variables $\phi^L$ and $\phi^R$ are not discrete, this event has probability 0, so the kernel of $W$ intersects the image of $V$ trivially and $X = VW$ has rank 2.

**Proof of Theorem 4.1 (Torus)**

Equation 8 shows $X = VW$ for the matrices $V, W$ defined by rows and columns respectively

$$V_{(a,b),:} = [\cos(\theta_a), \sin(\theta_a), \cos(\theta_b), \sin(\theta_b)] \tag{12}$$

$$W_{:,i} = \begin{bmatrix} \cos(\Phi_i^{L,a}) \\ -\sin(\Phi_i^{L,a}) \\ \cos(\Phi_i^{R,b}) \\ -\sin(\Phi_i^{R,b}) \end{bmatrix}. \tag{13}$$

The proof that $X$ has rank 4 is the same as the one for the respective statement in theorem 1 (uniformity of phases give the rank of $V$ and $W$ and the independence of the image of $V$ and kernel of $W$).

$\square$

## C  STATISTICAL SIGNIFICANCE OF MAIN RESULTS

### C.1  FIGURE 5

We trained 703 models of each architecture, being MLP vec add, Attention 0.0 and 1.0, and MLP concat, and recorded the locations of the max activations of all neurons across all $(a, b)$ inputs to the network. We also computed the center of mass of each neuron as this doesn't always align with the max preactivation (though it tends to be close).

| Description | MMD | p-value | Interpretation |
|---|---|---|---|
| MLP vec add vs Attention 0.0 | 0.0968 | 0.0000 | Moderate difference; highly significant |
| MLP vec add vs Attention 1.0 | 0.1239 | 0.0000 | Clear difference; highly significant |
| MLP vec add vs MLP concat | 0.2889 | 0.0000 | Very strong difference; highly significant |
| Attention 0.0 vs Attention 1.0 | 0.0338 | 0.0000 | Subtle difference; highly significant |
| Attention 0.0 vs MLP concat | 0.1987 | 0.0000 | Strong difference; highly significant |
| Attention 1.0 vs MLP concat | 0.1723 | 0.0000 | Strong difference; highly significant |

(a) Row 1: Max activation

| Description | MMD | p-value | Interpretation |
|---|---|---|---|
| MLP vec add vs Attention 0.0 | 0.0583 | 0.0000 | Small difference; highly significant |
| MLP vec add vs Attention 1.0 | 0.0689 | 0.0000 | Moderate difference; highly significant |
| MLP vec add vs MLP concat | 0.2614 | 0.0000 | Very strong difference; highly significant |
| Attention 0.0 vs Attention 1.0 | 0.0210 | 0.0084 | Subtle difference; highly significant |
| Attention 0.0 vs MLP concat | 0.2126 | 0.0000 | Strong difference; highly significant |
| Attention 1.0 vs MLP concat | 0.1947 | 0.0000 | Strong difference; highly significant |

(b) Row 2: Center of mass

Table 3: Gaussian-kernel Maximum Mean Discrepancies (MMD) Gretton et al. (2012) and permutation p-values between the empirical distributions shown in Figure 5. For each architecture comparison, we sampled 20,000 points from each empirical distribution (derived from histogram-based neuron statistics), then computed the unbiased Gaussian-kernel MMD with a bandwidth chosen via the pooled median heuristic. Significance was assessed using 50,000 permutation tests per comparison.

## D  TORUS DISTANCE METRIC

We introduce and study the *torus distance* metric.

Figure 7. We trained 703 models of each architecture with 512 neurons in its hidden layer (MLP vec add, Attention 0.0 and 1.0, and MLP concat), and recorded the $a, b$ value of where the max activation of a neuron takes place across all $(a, b)$ inputs to the network and all neurons. We also computed the $(a, b)$ values for the location of the center of mass of each neuron as this doesn't always align with the max preactivation (though it tends to be close). Then we compute the shortest torus distance from the point of the max activation or the center of mass, to the line $a = b$.

## E  PREVIOUS INTERPRETABILITY METRICS (ZHONG ET AL., 2023)

### E.1  DEFINITIONS OF GRADIENT SYMMETRICITY AND DISTANCE IRRELEVANCE

**Gradient symmetricity** measures, over some subset of input-output triples $(a, b, c)$, the average cosine similarity between the gradient of the output logit $Q_{(a,b,c)}$ with respect to the input embeddings of $a$ and $b$. For a network with embedding layer $\mathbf{E}$ and a set $S \subseteq \mathbb{Z}_p^3$ of input-output triples:

$$s_g = \frac{1}{|S|} \sum_{(a,b,c) \in S} \text{sim} \left( \frac{\partial Q_{abc}}{\partial \mathbf{E}_a}, \frac{\partial Q_{abc}}{\partial \mathbf{E}_b} \right)$$

where $\text{sim}(u, v) = \frac{u \cdot v}{\|u\|\|v\|}$ is the cosine similarity. It is evident that $s_g \in [-1, 1]$.

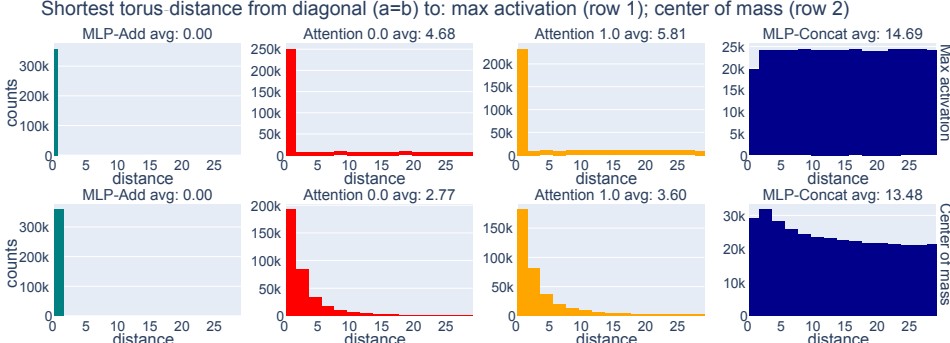

Figure 7: Histograms of torus-distance from each neuron's phase to the diagonal $a = b$, across 703 trained models. MLP-Add neurons align perfectly with the diagonal, Attention 0.0 and 1.0 show increasing off-diagonal spread, and MLP-Concat exhibits broadly distributed activations on the torus.

Table 4: Gaussian-kernel Maximum Mean Discrepancies (MMD) Gretton et al. (2012) and permutation p-values between the empirical distributions shown in Figure 7. For each architecture comparison, we sampled 2000 points from each empirical distribution (derived from histogram-based neuron statistics), then computed the unbiased Gaussian-kernel MMD with a bandwidth chosen via the pooled median heuristic. Significance was assessed using 5000 permutation tests per comparison.

(a) Row 1: Max activation

| Description | MMD | p-value | Interpretation |
|---|---|---|---|
| MLP vec add vs Attention 0.0 | 0.3032 | 0.0000 | Strong difference; highly significant |
| MLP vec add vs Attention 1.0 | 0.3888 | 0.0000 | Very strong difference; highly significant |
| MLP vec add vs MLP concat | 0.9508 | 0.0000 | Extremely strong difference; highly significant |
| Attention 0.0 vs Attention 1.0 | 0.0705 | 0.0000 | Moderate difference; highly significant |
| Attention 0.0 vs MLP concat | 0.6323 | 0.0000 | Very strong difference; highly significant |
| Attention 1.0 vs MLP concat | 0.5695 | 0.0000 | Very strong difference; highly significant |

(b) Row 2: Center of mass

| Description | MMD | p-value | Interpretation |
|---|---|---|---|
| MLP vec add vs Attention 0.0 | 0.7727 | 0.0000 | Extremely strong difference; highly significant |
| MLP vec add vs Attention 1.0 | 0.7517 | 0.0000 | Extremely strong difference; highly significant |
| MLP vec add vs MLP concat | 0.9148 | 0.0000 | Extremely strong difference; highly significant |
| Attention 0.0 vs Attention 1.0 | 0.0520 | 0.0006 | Moderate difference; highly significant |
| Attention 0.0 vs MLP concat | 0.7022 | 0.0000 | Very strong difference; highly significant |
| Attention 1.0 vs MLP concat | 0.6391 | 0.0000 | Very strong difference; highly significant |

**Distance irrelevance** quantifies how much the model's outputs depend on the distance between $a$ and $b$. For each distance $d$, we compute the standard deviation of correct logits over all $(a, b)$ pairs where $a - b = d$ and average over all distances. It's normalized by the standard deviation over all data.

Formally, let $L_{i,j} = Q_{ij,i+j}$ be the correct logit matrix. The distance irrelevance $q$ is defined as:

$$q = \frac{\frac{1}{p} \sum_{d \in \mathbb{Z}_p} \text{std}(\{L_{i,i+d} | i \in \mathbb{Z}_p\})}{\text{std}(\{L_{i,j} | i, j \in \mathbb{Z}_p\})}$$

where $q \in [0, 1]$, with higher values indicating greater irrelevance to input distance.

E.2    EVALUATING GRADIENT SYMMETRICITY AND DISTANCE IRRELEVANCE METRICS

Figure 8 shows the mean and standard deviation of the gradient symmetricity and distance irrelevance metrics from Zhong et al. (2023). Unlike Zhong et al. (2023), who report gradient symmetricity results over a randomly selected subset of 100 input-output triples $(a, b, c) \in \mathbb{Z}_p^3$, we compute

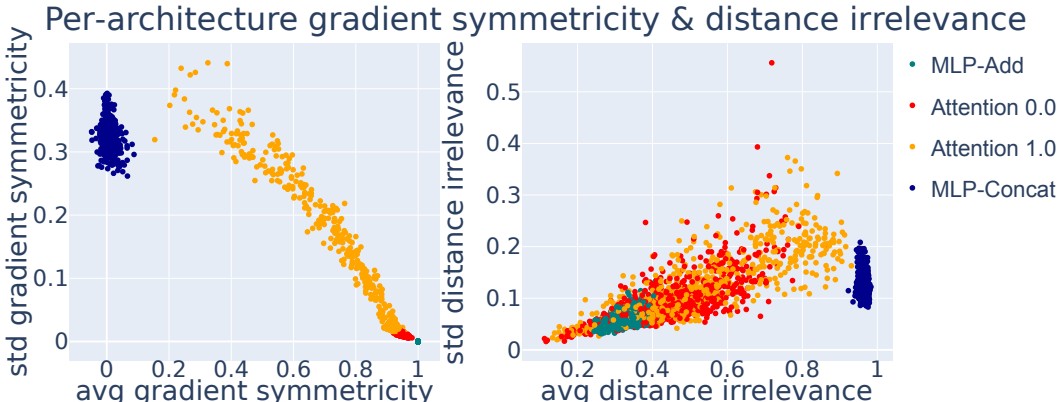

Figure 8: Evaluation of gradient symmetricity (left) and distance irrelevance (right). Each point shows the average (avg) and standard deviation (std) of one trained network. MLP-Add and MLP-Concat lie at nearly opposite extremes, while attention 0.0 and 1.0 overlap substantially. Gradient symmetricity separates Attention 1.0 better, but neither metric **always** distinguishes between Attention 1.0 and 0.0.

the metric exhaustively across **all** $59^3 = 205,370$ triples to add accuracy. See Appendix G for the GPU-optimized procedure.

MLP-Add and MLP-Concat cluster on opposite extremes, implying the metrics just identify whether neurons have phases $\phi^L \neq \phi^R$. MLP-Add models have high gradient symmetricity and low distance irrelevance and MLP-Concat models have low gradient symmetricity and high distance irrelevance. Attention 1.0 models span a wide range between these extremes depending on two factors: 1) how well the frequencies they learned intersect and 2) how well neurons are able to get their activation center of mass away from the $\phi^L = \phi^R$ line. Attention 0.0 is closer to MLP-Add than Attention 1.0 because it's harder for this architecture to learn $\phi^L \neq \phi^R$. Notably, failure cases exist using both: **neither metric distinguishes between Attention 1.0 and 0.0 models.**

MMD results for these two metrics are reported below, again showing that the distance between attention 0.0 and attention 1.0 models is small. This is the case even those these metrics were chosen to differentiate between the two architectures.

Using just the x-axis (since the y-axis on those plots is the std dev) MMD results are presented next.

We can conclude that the attention transformers are far from vector addition, and very close to each other under all metrics.

[h]

Table 5: Permutation–test MMDs on the empirical gradient symmetricity and distance irrelevance distributions across all architectures. All p-values are $\leq 10^{-6}$ (reported as 0.0000).

(a) Gradient symmetricity (2-D: *avg* and *std*)

| Description | MMD | p-value | Interpretation |
|---|---|---|---|
| MLP vec add vs Attention 0.0 | 1.2725 | 0.0000 | Extremely strong difference; highly significant |
| MLP vec add vs Attention 1.0 | 0.9688 | 0.0000 | Extremely strong difference; highly significant |
| MLP vec add vs MLP concat | 1.3471 | 0.0000 | Extremely strong difference; highly significant |
| Attention 0.0 vs Attention 1.0 | 0.7750 | 0.0000 | Very strong difference; highly significant |
| Attention 0.0 vs MLP concat | 1.3503 | 0.0000 | Extremely strong difference; highly significant |
| Attention 1.0 vs MLP concat | 1.2360 | 0.0000 | Extremely strong difference; highly significant |

(b) Distance irrelevance (2-D: *avg* and *std*)

| Description | MMD | p-value | Interpretation |
|---|---|---|---|
| MLP vec add vs Attention 0.0 | 0.7534 | 0.0000 | Very strong difference; highly significant |
| MLP vec add vs Attention 1.0 | 0.7079 | 0.0000 | Very strong difference; highly significant |
| MLP vec add vs MLP concat | 1.2488 | 0.0000 | Extremely strong difference; highly significant |
| Attention 0.0 vs Attention 1.0 | 0.2078 | 0.0000 | Moderate difference; highly significant |
| Attention 0.0 vs MLP concat | 1.2255 | 0.0000 | Extremely strong difference; highly significant |
| Attention 1.0 vs MLP concat | 1.0990 | 0.0000 | Extremely strong difference; highly significant |

[h]

Table 6: Permutation-test MMDs on scatter-plot averages only (1-D). All $p$–values are $\leq 10^{-6}$, so every difference is "highly significant." Note that the distance between attention 0.0, attention 1.0, and MLP vec add is large, implying they are not performing vector addition.

(a) Row 3: Gradient symmetricity (avg only)

| Description | MMD | $p$-value | Interpretation |
|---|---|---|---|
| MLP vec add vs Attention 0.0 | 1.2755 | 0.0000 | Extremely strong difference; highly significant |
| MLP vec add vs Attention 1.0 | 0.9842 | 0.0000 | Extremely strong difference; highly significant |
| MLP vec add vs MLP concat | 1.3833 | 0.0000 | Extremely strong difference; highly significant |
| Attention 0.0 vs Attention 1.0 | 0.7726 | 0.0000 | Extremely strong difference; highly significant |
| Attention 0.0 vs MLP concat | 1.3802 | 0.0000 | Extremely strong difference; highly significant |
| Attention 1.0 vs MLP concat | 1.2559 | 0.0000 | Extremely strong difference; highly significant |

(b) Row 4: Distance irrelevance (avg only)

| Description | MMD | $p$-value | Interpretation |
|---|---|---|---|
| MLP vec add vs Attention 0.0 | 0.7739 | 0.0000 | Extremely strong difference; highly significant |
| MLP vec add vs Attention 1.0 | 0.7268 | 0.0000 | Very strong difference; highly significant |
| MLP vec add vs MLP concat | 1.2501 | 0.0000 | Extremely strong difference; highly significant |
| Attention 0.0 vs Attention 1.0 | 0.2109 | 0.0000 | Strong difference; highly significant |
| Attention 0.0 vs MLP concat | 1.2443 | 0.0000 | Extremely strong difference; highly significant |
| Attention 1.0 vs MLP concat | 1.1093 | 0.0000 | Extremely strong difference; highly significant |

## F  ADDITIONAL COMMENTARY ON THE RESULTS

### F.1  THE ATTENTION 1.0 MODEL IS USING ATTENTION AS A WEAK NON-LINEARITY.

It's noteworthy that some neurons in the first MLP layer of transformers have learned one frequency $f$, and a sum of two different sinusoidal features (*i.e. a superposition via linear combination)*. These

are of one first-order sinusoidal feature (the simple neuron model) $\sin(\frac{2\pi fa}{n}) + \sin(\frac{2\pi fb}{n})$ and also a second term, being second order and $\cos(\frac{f(a+b-c)}{n})$, with both terms having the same $f$ value.

This is why the sum of neuron-cluster post-activations plot (Figure 3) has a bit of "off diagonal patchy-spread" that runs from the top left to the bottom right at 45 degrees in only the attention 1.0 model. These small cloudy patches are caused by a few neurons activating on $\cos(\frac{f(a+b-c)}{n})$.

Thus, it's the case that for modular addition, the attention layer and its sigmoidal non-linearity is being used like a weak non-linear layer. Sigmoidal attention isn't a strong enough non-linearity for the network to have all neurons with frequency $f$ learn the second order $\cos(\frac{f(a+b-c)}{n})$ that's typically seen in the second and third layers. Indeed, a past work, McCracken et al. (2025), utilized a rigorous empirical framework to inspect this over layers, learning rates, and l2 weight decay hyperparameters. They showed that across these settings, the best fit in the first layer in all neurons in transformers still comes from the simple neuron model.

## G   GPU-OPTIMIZED COMPUTATIONS

### G.1   GPU-OPTIMIZED CENTER-OF-MASS IN CIRCULAR COORDINATES

Let $p$ be the grid size and for each neuron $n = 1, \ldots, N$ we have a pre-activation map

$$x_{i,j}^{(n)}, \quad (i, j = 0, \ldots, p - 1).$$

Define nonnegative weights

$$w_{i,j}^{(n)} = \big| x_{i,j}^{(n)} \big|.$$

Let $f_n \in \{1, \ldots, \lfloor p/2 \rfloor\}$ be the dominant frequency for neuron $n$, and let

$$f_n^{-1} \text{ be the modular inverse of } f_n \text{ modulo } p, \qquad f_n\, f_n^{-1} \equiv 1 \pmod{p}.$$

Convert the row index $i$ and column index $j$ into angles ("un-wrapping" by $f_n^{-1}$):

$$\theta_i^{(n)} = \frac{2\pi}{p}\, f_n^{-1}\, i, \qquad \phi_j^{(n)} = \frac{2\pi}{p}\, f_n^{-1}\, j.$$

Form the two complex phasor sums

$$S_a^{(n)} = \sum_{i=0}^{p-1}\sum_{j=0}^{p-1} w_{i,j}^{(n)}\, \exp\!\big(i\,\theta_i^{(n)}\big),$$

$$S_b^{(n)} = \sum_{i=0}^{p-1}\sum_{j=0}^{p-1} w_{i,j}^{(n)}\, \exp\!\big(i\,\phi_j^{(n)}\big).$$

The arguments of these sums give the circular means of each axis:

$$\mu_a^{(n)} = \arg\!\big(S_a^{(n)}\big), \quad \mu_b^{(n)} = \arg\!\big(S_b^{(n)}\big),$$

where $\arg$ returns an angle in $(-\pi, \pi]$. To ensure a nonnegative result, normalize into $[0, 2\pi)$:

$$\mu^+ = \big(\mu + 2\pi\big) \bmod 2\pi.$$

Finally, map back from the angular domain to grid coordinates:

$$\mathrm{CoM}_a^{(n)} = \frac{p}{2\pi}\, \mu_a^{(n)+}, \qquad \mathrm{CoM}_b^{(n)} = \frac{p}{2\pi}\, \mu_b^{(n)+}.$$

This handles wrap-around at the boundaries automatically and weights each location $(i, j)$ by $|x_{i,j}^{(n)}|$, producing a smooth, circularly-aware center of mass. All tensor operations—angle computation, complex exponentials, and weighted sums—are expressed as parallel array primitives that JAX can JIT-compile and fuse into a single GPU kernel launch, eliminating Python-level overhead. By precomputing the angle grids and performing the phasor sums inside one jitted function, this implementation fully exploits GPU parallelism and memory coalescing for maximal throughput.

## G.2 GPU-VECTORIZED DISTANCE IRRELEVANCE OVER ALL $n^2$ INPUT PAIRS $(a, b)$

Let
$$\mathcal{I} = \big\{ (a,b) \mid a, b \in \{0, \ldots, n-1\} \big\},$$
and order its elements lexicographically:
$$X = \big[ (a_0, b_0), (a_1, b_1), \ldots, (a_{n^2-1}, b_{n^2-1}) \big] \in \mathbb{Z}^{n^2 \times 2}.$$

A $n^2$ single batched forward pass on the GPU computes
$$\text{Logits} = \text{Transformer}(X) \in \mathbb{R}^{n^2 \times n},$$
producing all $n^2 \cdot n$ output logits in parallel. We then extract the "correct-class" logit for each input:
$$y_k = \text{Logits}_{k, (a_k + b_k) \bmod n}, \qquad k = 0, \ldots, n^2 - 1.$$

Next we reshape $y$ into an $n \times n$ matrix $L$ by
$$L_{i,j} = y_k \quad \text{where} \quad i = (a_k + b_k) \bmod n, \; j = (a_k - b_k) \bmod n.$$

All of the above: embedding lookup, attention, MLP, softmax and the advanced indexing is implemented as two large vectorized kernels (the batched forward pass and the gather), so each of the $n^2$ inputs is handled in $O(1)$ time but fully in parallel on the GPU.

Finally, define
$$\sigma_{\text{global}} = \sqrt{\frac{1}{n^2} \sum_{i,j} \big( L_{i,j} - \mu \big)^2}, \quad \mu = \frac{1}{n^2} \sum_{i,j} L_{i,j},$$
and for each "distance" $j$
$$\sigma_j = \sqrt{\frac{1}{n} \sum_i \big( L_{i,j} - \bar{L}_{\cdot j} \big)^2}, \quad \bar{L}_{\cdot j} = \frac{1}{n} \sum_i L_{i,j}, \quad q_j = \frac{\sigma_j}{\sigma_{\text{global}}}.$$

We report
$$\bar{q} = \frac{1}{n} \sum_{j=0}^{n-1} q_j, \qquad \text{std}(q) = \sqrt{\frac{1}{n} \sum_{j=0}^{n-1} \big( q_j - \bar{q} \big)^2}.$$

## G.3 GPU-OPTIMIZED GRADIENT SYMMETRICITY OVER ALL $n^3$ TRIPLETS $(a, b, c)$

Let
$$E \in \mathbb{R}^{n \times d} \quad \text{with} \quad d = 128$$
be the learned embedding matrix, and denote by
$$Q\big( E_a, E_b \big)_c$$
the scalar logit for class $c$ obtained by feeding the pair of embeddings $(E_a, E_b)$ into the model. We define the per-triplet gradient cosine-similarity as
$$S(a, b, c) = \frac{\big\langle \nabla_{E_a} Q(E_a, E_b)_c, \; \nabla_{E_b} Q(E_a, E_b)_c \big\rangle}{\big\| \nabla_{E_a} Q(E_a, E_b)_c \big\| \, \big\| \nabla_{E_b} Q(E_a, E_b)_c \big\|},$$
for all $(a, b, c) \in \{0, \ldots, n-1\}^3$.

To compute $\{S(a, b, c)\}$ over the full $n^3$ grid in one fused GPU kernel, we first form three index tensors
$$A_{i,j,k} = i, \quad B_{i,j,k} = j, \quad C_{i,j,k} = k, \quad i, j, k = 0, \ldots, n-1,$$
then flatten to vectors $a = \text{vec}(A)$, $b = \text{vec}(B)$, $c = \text{vec}(C) \in \{0, \ldots, n-1\}^{n^3}$. We gather the embeddings
$$\text{emb}_a = E[a] \in \mathbb{R}^{n^3 \times d}, \quad \text{emb}_b = E[b] \in \mathbb{R}^{n^3 \times d},$$

and in JAX compute

$$\mathbf{g}_a \;=\; \mathrm{vmap}\big((e_a, e_b, c) \mapsto \nabla_{E_a} Q(e_a, e_b)_c\big)(\mathsf{emb}_a,\, \mathsf{emb}_b,\, c),$$

$$\mathbf{g}_b \;=\; \mathrm{vmap}\big((e_a, e_b, c) \mapsto \nabla_{E_b} Q(e_a, e_b)_c\big)(\mathsf{emb}_a,\, \mathsf{emb}_b,\, c),$$

each producing an $(n^3 \times d)$-shaped array. Finally the similarity vector is

$$\mathbf{S} \;=\; \frac{\mathbf{g}_a \odot \mathbf{g}_b}{\|\mathbf{g}_a\|\,\|\mathbf{g}_b\|} \;\in\; \mathbb{R}^{n^3},$$

and we report

$$\overline{S} = \frac{1}{n^3}\sum_{i=1}^{n^3} S_i, \qquad \sigma_S = \sqrt{\frac{1}{n^3}\sum_{i=1}^{n^3}\left(S_i - \overline{S}\right)^2}.$$

**Runtime.** Because we express $\mathbf{g}_a, \mathbf{g}_b$ and the subsequent dot-and-norm entirely inside a single `@jax.jit`+ `vmap` invocation, XLA lowers it to one GPU kernel that processes all $n^3$ triplets in parallel. The kernel dispatch cost is therefore $O(1)$, and each triplet's gradient and cosine computations are fused into vectorized instructions with constant per-element overhead. Although the total arithmetic work is $O(n^3)$, the full data-parallel execution means the wall-clock latency grows sub-linearly in $n^3$ and the per-triplet overhead remains effectively constant.

## H    HYPOTHESIS: MODULAR ADDITION AS A FACTORED MAP FROM THE TORUS TO THE CIRCLE

Modular addition is the function $\mathbb{Z}_n \times \mathbb{Z}_n \to \mathbb{Z}_n$ sending the pair $(a, b)$ to $c = a + b \mod n$. Geometrically, we may embed $a \in \mathbb{Z}_n$ on the unit circle $\mathbb{R}^2$ via $\mathbf{E}_a = (\cos(2\pi a/n), \sin(2\pi a/n))$. So the product space $\mathbb{Z}_n \times \mathbb{Z}_n$ embeds into $\mathbb{R}^4$ as a discretized torus, parameterized by

$$(a, b) \mapsto (\cos u, \sin u, \cos v, \sin v), \quad u = 2\pi a/n,\ v = 2\pi b/n.$$

In this embedding, modular addition corresponds to the following map from the torus to the circle:

$$(x_1, x_2, x_3, x_4) \mapsto (x_1 x_3 - x_2 x_4,\, x_1 x_4 + x_2 x_3).$$

Parameterizing by angles, this becomes the familiar trigonometric identity

$$(\cos u, \sin u, \cos v, \sin v) \mapsto (\cos(u + v), \sin(u + v)).$$

We claim that networks we study are approximating this specific geometric map from a torus in $\mathbb{R}^4$ to a circle in $\mathbb{R}^2$. In Section 3.1 we saw that the "clock" and "pizza" interpretations included learned embeddings of the form of $\mathbf{E}_a$. Taken together, these embeddings define a torus $\mathbf{T}^2$ as the input representation space. **Our hypothesis** is that architectures (MLP-Add, Attention 0.0 and 1.0, MLP-Concat) do not learn fundamentally different solutions; rather they factor the same torus-to-circle map via different intermediate representations. Fig. 4 shows factorizations of this map.

