# OpenReview forum: "On The Geometry and Topology of Representations: the Manifolds of Modular Addition"
_ICLR.cc/2026/Conference — ICLR 2026 Poster_

### Official Review · Reviewer_9zL4 · 2025-10-15

**Soundness:** 4
**Presentation:** 3
**Contribution:** 4
**Rating:** 8
**Confidence:** 3

**Summary:**

The paper investigates internal representations of networks trained to perform modular addition. Prior work claims that networks either learn a Clock or Pizza circuit. The paper shows that for a number of networks, the manifold at early layers is a torus which slowly morphs to be more circle-like in later layers (including a disk) although not completely a circle. This is shown first theoretically, then empirically through a number of analyses.

**Strengths:**

The paper's key strength is the depth of the empirical analysis. The authors consider a number of techniques to probe the representations learned by the networks including PCA, activation strengths, PAD and Betti numbers. This is all done for a number of different architectures. The theoretical analysis also seems sound and is convincing.

Overall, the paper would likely be of significance to those in mechanistic interpretability, particularly those studying modular addition.

**Weaknesses:**

In my view, the paper doesn't have many weaknesses. My main critique is on clarity: sections 4.1 and 5 could benefit from more intuition provided for the theorem and proposed analyses. For instance, it's not immediately obvious what the factorization structure of X in Theorem 4.1 has to do with geometry. Similarly, in the description of PAD, the significance of a strong diagonal is never explicitly explained. I would encourage the authors to use their extra page in the revision to more slowly walk through these harder-to-understand areas of the paper.

Minor points:
- \mathbf Z and \mathbb Z appear to be used inconsistently in places (as well as for T in the appendix)
- It's unclear why bold font is used for E and Z but not for any other symbols
- The color scheme for visualizing numbers from 0-58 is not cyclic (it looks like viridis); I recommend using a cyclic color scheme
- Figure 6 text is a bit small, especially for the legend

**Questions:**

- Please clarify sections 4.1 and 5
- See minor points above

---

> ### Author Response · Authors · 2025-11-28
> **Official Comment by Authors**
>
> Thank you for reviewing our work and for recognizing the depth of our empirical analysis and the soundness of our theoretical analysis!
>
> ### Weaknesses:
>
> **Theorem 4.1:** This theorem essentially tells us that our data matrix (the frequency-specific representations) while living in a very high-dimensional space (given by the number of neurons in a cluster) has a low-dimensional manifold structure: either a 2d disc if all phases are on the diagonal or a 4d torus if phases are allowed to vary arbitrarily off the diagonal. The factorization structure of X in Theorem 4.1 essentially tells us that we can decompose our data matrix into a low-dimensional manifold (V gives us the 2d or 4d coordinates of our manifold) and a linear embedding of this manifold into a higher-dimensional space of dimension equal to the number of neurons in our frequency cluster (W gives us this embedding into the higher dimensional space).
>
> **PAD and significance of strong diagonal:** The strong diagonal relates to phase concentration on the diagonal which results in a vector addition disc. It provides a concise way to predict the structure of the representation manifold; e.g. by Theorem 4.1, when the PAD is concentrated on the diagonal, we know that the representation manifold will be a disc; alternatively, when the PAD is uniform, the torus emerges.
>
> **Regarding the minor points:** thanks. We are going to unify the \mathbb and \mathbf notation as you suggested, increase the font size of labels/legends in the figures, and experiment with cyclic color schemes in our revision.
>
> Thanks again for highlighting some areas where clarity can be improved, we believe adding the clarifications as discussed here will strengthen our work.

---

### Official Review · Reviewer_3muT · 2025-10-23

**Soundness:** 3
**Presentation:** 3
**Contribution:** 2
**Rating:** 4
**Confidence:** 3

**Summary:**

This paper revisits the modular addition benchmark, showing that models previously thought to learn distinct “Clock” and “Pizza” circuits in fact converge to geometrically and topologically equivalent manifolds. Using the simple neuron model (from McCracken et al., 2025), the authors derive closed-form expressions for layer-1 preactivations and prove that under mild assumptions on phase distributions, all networks learn either a disc (vector-addition) or torus manifold. They then verify these findings empirically via Phase Alignment Distributions (PAD) and Betti number analyses across hundreds of networks, concluding that the supposed circuit diversity of Zhong et al. (2023) arises from superficial differences.
The work further argues that the universality hypothesis—that similar networks learn similar circuits—remains valid, thus unifying previously divergent mechanistic interpretations of modular arithmetic.

**Strengths:**

Theoretical clarity: Derivation of closed-form preactivation manifolds (Theorem 4.1) provides a clear and elegant link between Fourier structure and representation geometry.

Unified interpretation: The paper offers a coherent view showing that previously “distinct” circuits (Clock vs Pizza) are mathematically equivalent, reinforcing the universality hypothesis.

Methodological novelty: Introduces quantitative topology tools (Betti distributions, PAD + MMD metrics) for large-scale circuit comparison.
Empirical rigor: Evaluates 703 trained models across multiple architectures, giving the conclusions strong empirical grounding.

Conceptual impact: Connects manifold geometry, phase statistics, and interpretability—potentially inspiring similar analyses for other arithmetic circuits.

**Weaknesses:**

Missing connection to real LLMs: https://arxiv.org/abs/2406.03445 shows that similar Fourier-modular features emerge in large pretrained transformers, yet this paper does not discuss whether such real representations share the same torus or disc topology. Integrating this perspective could help demonstrate broader relevance beyond synthetic tasks.

Insufficient engagement with prior findings: https://arxiv.org/pdf/2311.07568 and https://arxiv.org/pdf/2402.09469 already show that linear superposition of $\cos a$ and $\cos b$ can yield $\cos(a+b)$ without attention and also characterize attention outputs in Fourier space. A deeper comparison or reconciliation with these works would make the analysis more convincing.

Limited methodological novelty: The TDA methods (Betti numbers, persistent homology) are established tools; the contribution mainly lies in their application to this particular setting.

Questionable realism: The MLP-Concat baseline, while illustrative, trivially produces a torus and is rarely used in realistic architectures. Clarifying its motivation would strengthen the empirical design.

**Questions:**

How does your explanation differ from https://arxiv.org/pdf/2311.07568, which showed that $\cos(2\pi f(a+b))$ can emerge from linear mixing of $\cos a$ and $\cos b$ with activation function? Maybe activation function in real model can also get  $\cos(2\pi f(a+b))$. Hence it may not contradict to the clock algorithm.

In https://arxiv.org/pdf/2402.09469, attention outputs are sparse in Fourier space—does this align with your observations? Are the attention outputs closer to $\cos a + \cos b$ or to $\cos(a+b)$?

Have you examined whether similar disc or torus topologies appear in pretrained language models, such as those analyzed in https://arxiv.org/abs/2406.03445?

Beyond applying standard Betti-number analysis, what new theoretical insight or interpretability perspective does your TDA pipeline provide?

What motivates including MLP-Concat, given that it trivially yields a torus (four degrees of freedom) and is not common in modern architectures?

---

> ### Author Response · Authors · 2025-11-27
> **Official Comment by Authors**
>
> We’re very happy that you saw how our solution cleanly unifies the many theoretical and empirical interpretations in the current literature: “Theoretical clarity: Derivation of closed-form preactivation manifolds (Theorem 4.1) provides a clear and elegant link between Fourier structure and representation geometry”.
>
>
> > **W1.** Missing connection to real LLMs: https://arxiv.org/abs/2406.03445 shows that similar Fourier-modular features emerge in large pretrained transformers, yet this paper does not discuss whether such real representations share the same torus or disc topology. Integrating this perspective could help demonstrate broader relevance beyond synthetic tasks.
>
> Thank you for pointing out this cool research by Zhou et al., we’ve added a citation, however the study of LLMs is outside of our scope. Our experiments are deliberately restricted to our MLP architectures and the exact transformer architectures of prior work we describe in our paper. We train our own models and download trained models from prior work, where we show our theorem predicts and fully characterizes the empirical observations we make about the representation geometry and topology. That said, the observation of Fourier features in LLMs is definitely interesting and understanding their representations and operations does have great value given their widespread use. We feel however, this line of inquiry would be best supported by a paper of its own scoped to LLMs.
>
> > **W2.** Insufficient engagement with prior findings: https://arxiv.org/pdf/2311.07568 and https://arxiv.org/pdf/2402.09469 already show that linear superposition of $\cos a$ and $\cos b$  can yield $\cos(a+b)$ without attention and also characterize attention outputs in Fourier space. A deeper comparison or reconciliation with these works would make the analysis more convincing.
>
> Indeed, several works we cited (Gromov, Morwani et al., McCracken et al.) show that linear superposition of $\cos a$ and $\cos b$ (i.e. the simple neuron model) can yield $\cos(a+b)$ without attention. Thank you for the new reference to https://arxiv.org/pdf/2402.09469 (Li et al.) which provides further support of the simple neuron model, which we now cite in the manuscript.
>
> Our work is in agreement with these findings, and builds on them by identifying and fully characterizing the neural representations DNNs distribute over many neurons. Furthermore, our goal is to identify which architectures learn which neural representations, and learn about the nature of universality.
>
>
> > **W3.**  Limited methodological novelty: The TDA methods (Betti numbers, persistent homology) are established tools; the contribution mainly lies in their application to this particular setting.
>
>
> Our novel methodology allows us to directly interpret each neural representation and its corresponding activation geometry in deep neural networks trained on group tasks. We achieve this by identifying which neurons belong to each neural representation. This allows us to build the preactivation matrix for the entire neural representation and consequentially, directly inspect the manifolds of modular addition for the first time. We agree that TDA is an established tool in data science. In fact, we view it as a strength of our methodology that well-understood methods from TDA and geometry can be applied to obtain new theoretical and interpretative insight into modular addition representations and circuits. This contrasts with prior work on modular addition that had to infer the existence of different circuits via bespoke metrics.
>
> > **W4.** Questionable realism: The MLP-Concat baseline, while illustrative, trivially produces a torus and is rarely used in realistic architectures. Clarifying its motivation would strengthen the empirical design.
>
>
> Indeed, the MLP-Concat architecture is exotic, but this is exactly what makes it so useful. Our goal is to learn more about the possible representations of networks and what is common amongst them, and more broadly, we seek to find the appropriate formulation of the universality hypothesis. This means that we will not only consider models that have proven their usefulness in achieving state of the art results, but also architectures like MLP-Concat, which especially deserves mention as it learns a novel representation no prior work has reported. MLP-Concat does produce a torus representation, but that is what we were searching for since our Theorem predicts such neural representations can be learned, and no prior work had yet reported one. Furthermore, the fact MLP-Concat learns a torus is incredibly important due to its potential to affect discussions on the universality hypothesis. This is because we prove the pizzas learned by transformers are linear projections of the torus, thus, despite our work reporting different architectures can learn different neural representations, we’re able to conjecture “DNNs will learn a universal manifold to fit the data, or linear projections of it”.

---

> ### Author Response · Authors · 2025-11-27
> **Official Comment by Authors**
>
> > **Q1.** How does your explanation differ from https://arxiv.org/pdf/2311.07568, which showed that $\cos(2\pi f (a+b))$ can emerge from linear mixing of $\cos a$ and $\cos b$ with activation function? Maybe activation function in real model can also get $\cos(2\pi f(a+b))$. Hence it may not contradict to the clock algorithm.
>
>
> This is a good question. The clock algorithm of Zhong et al. states that the first intermediate representation (after attention and before the ReLU non-linearity of layer 1) has learned has a circle structure (Fig. 1). Our work shows that this is never the case, and that the circle structure only ever appears after ReLU and a subsequent linear projection onto the neurons of the next layer (or the logits, depending on the number of layers  ( Fig. 4 or Fig. 5)). This does not contradict Morwani et al., which shows that clocks can appear after a nonlinearity (e.g., later than the first hidden representation). Our work differs from Morwani by providing a complete characterization of the representation manifolds across layers under much more general assumptions.
>
>
> > **Q2.**  In https://arxiv.org/pdf/2402.09469, attention outputs are sparse in Fourier space—does this align with your observations? Are the attention outputs closer to $\cos a+\cos b$  or to $\cos(a+b)$?
>
> Our findings agree with Li et al.'s (https://arxiv.org/pdf/2402.09469) finding that the attention outputs are sparse in Fourier space (meaning they concentrate on one key frequency). The outputs of the attention are much closer to $\cos a + \cos b$, mean $R^2$ of fitting $\cos a + \cos b$ is $> 0.93$, but fitting $\cos(a+b)$ gives mean $R^2 < 0.03$.
>
>
> > **Q3.** Have you examined whether similar disc or torus topologies appear in pretrained language models, such as those analyzed in https://arxiv.org/abs/2406.03445?
>
> See response to weakness 1.
>
>
> > **Q4.** Beyond applying standard Betti-number analysis, what new theoretical insight or interpretability perspective does your TDA pipeline provide?
>
> See response to weakness 3.
>
> > **Q5.** What motivates including MLP-Concat, given that it trivially yields a torus (four degrees of freedom) and is not common in modern architectures?
>
> See response to weakness 4.
>
> Thank you again for your review. We hope our clarifications and answers to your questions are sufficient and that you’ll consider increasing your score. If anything remains unclear, we’re happy to make further clarifications.

---

### Official Review · Reviewer_yR58 · 2025-10-27

**Soundness:** 2
**Presentation:** 1
**Contribution:** 2
**Rating:** 2
**Confidence:** 4

**Summary:**

This work proposes a topological approach to interpretability in modular addition. Under the assumption of simple neurons, it is theoretically established that activations lie on a disc (when phases are in sync) or on a torus (when phases might be independent). The authors then perform experiments to shed light on the geometry of the activations. Based on their empirical analysis, they argue that previously identified mechanisms Pizza and Clock are in fact the same algorithm, restoring the universality hypothesis.

**Strengths:**

- The question of universal representations is important and modular addition is a good setup to study this question.
- The topological/geometric approach is interesing and sheds light on the representations learned by neural networks.
- The paper studies a diverse and comprehensive set of four architectures: MLP-add, MLP-concat, Attention 0.0 (transformers with constant attention), Attention 1.0 (transformers with learnable attention).
- The paper combines theoretical analysis with a comprehensive suite of experiments.
- The theoretical analysis appears correct (although the assumptions are up for debate).

**Weaknesses:**

1. The paper uses ambiguous language that might lead the reader to confuse neural network architectures and the mechanisms learned by these networks. For example, in lines 116-118, the terms Clock and Pizza are introduced as two different architectures, when in fact they are different mechanisms learnable by the same architecture (the standard transformer block: MLP + Softmax Attention). In lines 151-154, the authors are making a statement seemingly about archictures, when in reality they are refering to specific learned networks. I would suggest that the authors renouce the term "architecture" in favour of just "network", while clarifying that they are refereing to a specific mechanism learned by a specific architecture.
2. The numbers in the terms "Attention 0.0" and "Attention 1.0" probably reffer to the $\alpha$ constant introducted by Zhong et al. (2023), but this is not described here. I would recommend renaming "Attention 0" and "Attention 1" to "Constant-Attention" (or "Uniform-Attention") and "Learnable Attention", respectively.
3. The use of the letter E in eqs 2 and 3 is confusing by suggesting an embedding vector, or at least an activation, but after reading the paper several times this appears to be just a high level explanation (at least for the clock).
4. Most importantly, the idea that the clock algorithm can be reduced to the same "pizza disk" is very problematic. The Clock algorithm uses the attention mechanism to perform trigonometric operations. However, these operations require a linear attention regime (not softmax). Therefore, it is expected that most attention scores would concentrate closely around zero, where the application of softmax is approximately linear. However, this also implies that the output of attention would be approximately an average sum (as in the pizza), with the entire clock mechanism is encoded in the variance from this average. Therefore, it is not surprising that a PCA of the Clock reveals a pizza disk. However, a closer look at the plots reveals that the Clock PCA is not identical to the Pizza PCA: there is a lot of "noise" (hidden information) in the Clock plot. The same can be said about the sum of post-relu activations: we can see in Figure 3 that Attention 1.0 contains an additional (albeit weak) signal that is not present in the Attention 0.0. It is exactly this faint signal that is essential to the Clock algorithm. I strongly suspect that all of these apparant similarities would vanish if we considered the activation vectors projected into the row-space of the unembedding matrix.
5. The claim that "the Pizza and the Clock algorithms are actually the same algorithm" is **misleading**. Setting aside the weaknesses of the empirical analysis, the pizza and the clock are well understood mechanisms that are known to perform different computations. This papers reads to me like an argument that *quick sort* and *merge sort* are actually the same algorithm because they share some similarities. I believe this claim is greatly weakening this paper. The paper would be more interesting if the topological approach was used instead to shed light on the similarities and differences of the clock and the pizza. For example, can we characterize the complete topology of the clock algorithm beyond simple neurons?

**Questions:**

1. What exactly is meant by pre-activations at layer1? Is it the vectors from eqs 2 and 3?
2. Exactly what amount of the variance is described by the top-2 principal components for each network?
3. What do the plots looklike if we were to study the activations projected to the row-space of the unembedding matrix?
4. What do the 3rd and 4th principal components look like?
5. How much do the principal components 1-4 align with the unembedding matrix?
6. Does the torus topology of MLP-cat undermine the universality hypothesis?
7. Are the authors willing to tone down the claim that "the clock and pizza are the same algorithm", for example, to something like "topological similarities between the clock and the pizza"?

---

> ### Author Response · Authors · 2025-11-26
>
> Thank you for your review.
>
> We see two primary concerns in your review. 1. You’re concerned we claim two different circuits are the same circuit. 2. You’re concerned with the validity of the simple neuron assumption, that all neurons in layer 1 have first-order sinusoids as their preactivations.
>
> **Concern 1**. We apologize, there was some ambiguity between established models for circuits and which circuits are learned by which architectures in the paper. We claim both uniform and learnable attention architectures learn the same circuits, not disparate/different ones. Please note Fig. 1, shows readers the dataset manifolds corresponding to the clock and pizza circuits are totally different, by showing the topology (clocks have one hole, pizzas have zero) and showing the metric geometry (how points on each manifold are colored and the respective distances between points) are totally different between clocks and pizzas.
>
> “The theoretical analysis appears correct (although the assumptions are up for debate).”
>
> Context: We provide rigorous mathematical proof of Theorem 4.1, which states that clocks aren’t learned in layer 1 by any model that learns first-order sinusoidal neurons in the 1st layer.
>
> **Concern 2**. Your second concern seems to be with the simple neuron assumption, that neuron preactivations in the first layer come from first-order sinusoidal functions. This has been empirically confirmed by many prior works, either qualitatively, quantitatively, or both, at least four independent times. Nanda et al., qualitatively showed layer 1 neuron preactivations are first order in their Figure 12. Gromov showed this qualitatively in their Figure 2, and Morwani et al. showed it qualitatively in their Figure 1. Putting doubt to rest, McCracken et al. showed universality across three architectures: MLPs, and the uniform and learnable attention transformers of Zhong et al., by quantitatively finding first-order sinusoids (simple neurons) were the best fit to every architectures neurons in layer 1 (but not later layers). McCracken et al. did this via exhaustive search over training parameters, depths, widths, learning rates, weight decays and architectures (MLPs, transformers) every model trained learned first-order sinusoids (simple neurons) in the first layer.
>
> We want to show this beyond any possible doubt. Thus, we downloaded Nanda et al.’s python notebook and added fits for first-layer neuron preactivations of their paper’s mainline model. We find that first-order sinusoids fit with mean R^2=0.964 and second-order sinusoids of type cos(a+b) or cos(a-b) fit with mean R^2=0.0200. We also did this for Zhong et al. Model A (first-order mean R^2 0.998, second-order mean R^2 0.000) and Model B (first-order fits mean R^2 = 0.937, second-order mean R^2 = 0.0349). We added the notebooks to supplementary.zip. Thus, this assumption holds in all prior work and our present work.
>
> The second assumption is that the phases of the first-order sinusoidal functions are distributed uniformly, and Figure 5 demonstrates that the phases are distributed approximately uniformly over 4x703 seeds.
>
> Altogether, for these reasons, we believe our assumptions are very well justified.
>
> > Weakness 1. The paper uses ambiguous language that might lead the reader to confuse neural network architectures and the mechanisms learned by these networks. For example, in lines 116-118, the terms Clock and Pizza are introduced as two different architectures, when in fact they are different mechanisms learnable by the same architecture (the standard transformer block: MLP + Softmax Attention). In lines 151-154, the authors are making a statement seemingly about archictures, when in reality they are refering to specific learned networks. I would suggest that the authors renouce the term "architecture" in favour of just "network", while clarifying that they are refereing to a specific mechanism learned by a specific architecture.
>
> Zhong et al., claimed that transformers with fixed (uniform) attention always learn pizza circuits in layer 1, and that transformers with learnable attention always learn clock circuits in layer 1. This is why we refer to uniform attention (0.0) models as the models associated with learning pizzas and learnable attention (1.0) models as the models associated with learning clocks.
>
> We see how this could be confusing, since one of our main results is that architectures prior work claimed to learn clocks actually learn pizzas. We have made this explicit and have integrated your suggestions and hope the minor changes needed to address this are sufficient.

---

> ### Author Response · Authors · 2025-11-26
>
> > Weakness 2. The numbers in the terms "Attention 0.0" and "Attention 1.0" probably reffer to the  constant introducted by Zhong et al. (2023), but this is not described here. I would recommend renaming "Attention 0" and "Attention 1" to "Constant-Attention" (or "Uniform-Attention") and "Learnable Attention", respectively.
>
> See 1 above.
>
> > Weakness 3. The use of the letter E in eqs 2 and 3 is confusing by suggesting an embedding vector, or at least an activation, but after reading the paper several times this appears to be just a high level explanation (at least for the clock).
>
> This is the same notation Zhong et al. used, where the intention is that for each frequency learned by the network, there is a corresponding circuit with that frequency. Some of the elements in the embedding vectors therefore correspond to that frequency, and thus an abstraction for writing the part of the embedding vector associated with each frequency is the way we wrote it for E_a and E_b. Furthermore, Zhong et al. use this abstraction again for E_{ab}, which is now representing the preactivations after attention combines E_a and E_b. We have made this clear in the revision and added that it’s an abstraction we inherit from Zhong et al.
>
> > Weakness 4. Most importantly, the idea that the clock algorithm can be reduced to the same "pizza disk" is very problematic. The Clock algorithm uses the attention mechanism to perform trigonometric operations. However, these operations require a linear attention regime (not softmax). Therefore, it is expected that most attention scores would concentrate closely around zero, where the application of softmax is approximately linear. However, this also implies that the output of attention would be approximately an average sum (as in the pizza), with the entire clock mechanism is encoded in the variance from this average. Therefore, it is not surprising that a PCA of the Clock reveals a pizza d isk. However, a closer look at the plots reveals that the Clock PCA is not identical to the Pizza PCA: there is a lot of "noise" (hidden information) in the Clock plot. The same can be said about the sum of post-relu activations: we can see in Figure 3 that Attention 1.0 contains an additional (albeit weak) signal that is not present in the Attention 0.0. It is exactly this faint signal that is essential to the Clock algorithm. I strongly suspect that all of these apparant similarities would vanish if we considered the activation vectors projected into the row-space of the unembedding matrix.
>
> We appreciate the reviewers’ thoughtful suggestion to explore the structure of the noise in Clock PCA plots. We have added results showing the noise is only 5% of the variance explained below. Firstly, the table below shows that the mean variance explained by the first two principle components is >= 95%. Beyond that, we can state that the nature of the noise was quantitatively investigated by the main paper via persistent homology, which deterministically found no holes in the 4D object composed of the first four principle components over 703x2 random seeds. As Figure 1 shows, a clock necessarily must have one hole, and persistent homology found neither uniform or trainable attention transformers learn neural representations with one hole in layer 1. This is enough to rule out Zhong et al.’s claim that clock circuits are learned in layer 1.
>
> Thus, while the reviewer’s suggestion is a fair one, our evidence demonstrates that, in fact, the Clock algorithm never emerges in layer 1, as we originally claimed.
>
> ### Table of principal component variance explained of neural representations
>
> | PC | Layer 1 Preacts |  |  |  | Logits |  |  |  |
> | --- | --- | --- | --- | --- | --- | --- | --- | --- |
> | PC | MLP-Add | Uniform Attention | Trainable Attention | MLP-Concat | MLP-Add | Uniform Attention | Trainable Attention | MLP-Concat |
> | PC1 | 0.495 ± 0.026 | 0.501 ± 0.024 | 0.512 ± 0.027 | 0.252 ± 0.006 | 0.487 ± 0.039 | 0.515 ± 0.051 | 0.511 ± 0.018 | 0.502 ± 0.007 |
> | PC2 | 0.489 ± 0.025 | 0.484 ± 0.028 | 0.462 ± 0.034 | 0.250 ± 0.005 | 0.480 ± 0.040 | 0.482 ± 0.052 | 0.484 ± 0.018 | 0.494 ± 0.007 |
> | PC3 | 0.006 ± 0.004 | 0.007 ± 0.008 | 0.013 ± 0.009 | 0.246 ± 0.006 | 0.016 ± 0.035 | 0.002 ± 0.006 | 0.004 ± 0.006 | 0.002 ± 0.001 |
> | PC4 | 0.006 ± 0.004 | 0.005 ± 0.004 | 0.009 ± 0.007 | 0.243 ± 0.009 | 0.015 ± 0.034 | 0.001 ± 0.002 | 0.001 ± 0.002 | 0.001 ± 0.001 |

---

> > ### Author Response · Authors · 2025-11-26
> >
> > > Weakness 5. “The claim that "the Pizza and the Clock algorithms are actually the same algorithm" is misleading. Setting aside the weaknesses of the empirical analysis, the pizza and the clock are well understood mechanisms that are known to perform different computations. This papers reads to me like an argument that quick sort and merge sort are actually the same algorithm because they share some similarities. I believe this claim is greatly weakening this paper. The paper would be more interesting if the topological approach was used instead to shed light on the similarities and differences of the clock and the pizza. For example, can we characterize the complete topology of the clock algorithm beyond simple neurons?”
> >
> > Firstly, please see concern 1.
> >
> > Secondly, in response to your review, and the fact that persistent homology can only say that clocks aren’t learned in layer 1, we’ve decided to add two new quantitative tests that measure the similarity of each neural representation’s preactivation matrix with the ground truth manifolds predicted by Theorem 1.
> >
> > Since our method directly extracts every neuron associated with each neural representation, and builds each neural representation’s preactivation matrix in each layer, this is easy to do. We test the centered kernel distance (CKA) (see “Similarity of Neural Network Representations Revisited”, https://arxiv.org/abs/1905.00414), which measures how similarly two matrices represent the same data by comparing their pairwise feature-similarity patterns using a normalized kernel-based score. We also add representational similarity matrix (RSM) (Representational Similarity Analysis – Connecting the Branches of Systems Neuroscience https://pmc.ncbi.nlm.nih.gov/articles/PMC2605405/), we construct an RSM by computing the pairwise Euclidean distances between all datapoints on the manifold, and we construct an analogous distance matrix for the ground-truth representation. We then compute their Pearson correlation, which quantifies how similar the two representational geometries are. Both of these tests find our claims in the main paper are correct.
> >
> >
> > ### Layer 1 representations (CKA / RSM)
> >
> >
> > | Model               | vector_addition CKA | vector_addition RSM | torus CKA        | torus RSM         | clock CKA              | clock RSM         |
> > |---------------------|---------------------|----------------------|------------------|-------------------|------------------------|-------------------|
> > | one_embed           | 0.707 ± 0.012       | 0.578 ± 0.015        | 0.994 ± 0.011    | 0.991 ± 0.016     | 1.60e-11 ± 2.12e-10    | 0.112 ± 0.003     |
> > | one_embed_cheating  | 0.998 ± 0.017       | 0.998 ± 0.018        | 0.706 ± 0.012    | 0.579 ± 0.010     | -3.84e-12 ± 3.00e-10   | 0.109 ± 0.002     |
> > | transformer_0.0     | 0.988 ± 0.041       | 0.986 ± 0.044        | 0.699 ± 0.029    | 0.576 ± 0.022     | -1.01e-10 ± 7.60e-10   | 0.109 ± 0.004     |
> > | transformer_1.0     | 0.974 ± 0.028       | 0.972 ± 0.034        | 0.689 ± 0.020    | 0.581 ± 0.014     | 0.012 ± 0.010          | 0.126 ± 0.014     |
> >
> >
> > ### Cluster contributions to logits (CKA / RSM)
> >
> >
> > | Model               | clock CKA         | clock RSM         | torus CKA         | torus RSM         | vector_addition CKA | vector_addition RSM |
> > |---------------------|-------------------|-------------------|-------------------|-------------------|----------------------|----------------------|
> > | MLP-Concat           | 0.986 ± 0.025     | 0.981 ± 0.033     | 0.001 ± 0.001     | 0.115 ± 0.003     | 4.61e-04 ± 0.001     | 0.110 ± 0.006        |
> > | MLP-Add  | 0.926 ± 0.064     | 0.881 ± 0.097     | 0.026 ± 0.066     | 0.139 ± 0.056     | 0.037 ± 0.094        | 0.300 ± 0.148        |
> > | transformer_0.0     | 0.940 ± 0.071     | 0.917 ± 0.086     | 0.001 ± 0.003     | 0.115 ± 0.007     | 0.002 ± 0.005        | 0.258 ± 0.047        |
> > | transformer_1.0     | 0.941 ± 0.078     | 0.925 ± 0.091     | 0.002 ± 0.003     | 0.115 ± 0.008     | 0.002 ± 0.004        | 0.209 ± 0.060        |
> >
> >
> >
> >
> > To summarize, our novel methodology directly extracts the matrix corresponding to the activations of each neural representation in trained networks. With this matrix, it becomes easy to run any quantitative or qualitative tests that could be desired. Prior work didn’t have access to this, and had to instead infer their claims because they couldn’t directly show them.
> >
> > We’ve now provided four different primary quantitative experiments. Every quantitative experiment, each with over 1000 total seeds, shows Zhong et al.’s claim of “two stories in mechanistic interpretability” involving clocks or pizzas being learned in layer 1 to be false. Additionally, the new quantitative experiment showing the % variance explained by the neural representations matches our theorem’s prediction that the neural representations are rank 4 (torus) or rank 2 projections of the torus.

---

> > > ### Author Response · Authors · 2025-11-26
> > >
> > > Questions:
> > >
> > > >1. What exactly is meant by pre-activations at layer1? Is it the vectors from eqs 2 and 3?
> > >
> > > Yes, equations 2 and 3 are the form of what Zhong et al. claim the preactivations in layer 1 will be of the clock and pizza neural representations respectively.
> > >
> > > To be specific, we explain how we construct the preactivations of each neural representation on lines 238-244 of the original submission. We define layer neuron-cluster of preactivation matrices: “The cluster of preactivations of all neurons with key frequency f is the n^2 × |cluster f| matrix, made by flattening each neurons preactivation matrix and stacking the resulting vector for every neuron with the same key frequency. We call this matrix the neuron-cluster of preactivations matrix.”
> > >
> > > > 2. Exactly what amount of the variance is described by the top-2 principal components for each network?
> > >
> > > See the table in weakness 4, which shows both transformers and MLP-Add learn neural representations corresponding to rank 2 manifolds in each layer (and at the logits).
> > >
> > >
> > > > 3. What do the plots looklike if we were to study the activations projected to the row-space of the unembedding matrix?
> > >
> > > They look the same: model’s qualitatively show homology equivalent to clocks at this stage (note the hole). They’ve also adopted a coloring that’s similar to the clock. See https://postimg.cc/w1wwyqnr which also shows the CKA scores between the clock and pizza circuits ground truth manifolds in the title of the subplots. Note that the CKA scores here are computed on the 2D PCAs corresponding to the plotted points. Plese note: in the table above we used the full matrix (not an approximation).
> > >
> > > > 4. What do the 3rd and 4th principal components look like?
> > >
> > > See this plot visualizing PC0 vs PC1 and PC2 vs PC3, which shows that the 3rd and 4th principal components in layer 1 are qualitatively equivalent between the two transformers. https://postimg.cc/xcPbcy0s.
> > >
> > > > 5. How much do the principal components 1-4 align with the unembedding matrix?
> > >
> > > Can you please clarify what you mean by this? Our methodology tells us the exact manifold the activation geometry of each neural representation takes place on both before and after applying the unembedding matrix. The cumulative “alignment” from the first four principal components is >99% variance explained, if that’s what you mean, but again, we have the full matrix, so we don’t need to use PCA to compute quantitative tests other than our qualitative visualizations.
> > >
> > > > 6. Does the torus topology of MLP-cat undermine the universality hypothesis?
> > >
> > > This is a good question, and indeed our results probed us to rethink how to interpret the notion of universality.
> > > MLP-Concat (unlike the other architectures) is provided more information about the individual inputs because it doesn’t add the embedding vectors together and is thus able to reconstruct the 4d torus. In the other architectures, attention adds the embedding vectors together instead of concatenating them. Since there’s only one embedding matrix, this means that for all datapoints (a,b) where a=b, the same vector is used twice, and after they’re added together (producing one vector), it’s impossible to distinguish what came from E_a and E_b. Thus, it makes sense these architectures learn a disc, and this is why we created MLP-Add to demonstrate we could make an MLP that learned the exact same manifold as the transformers.
> > >
> > > Now, our theorem gives the closed-form equations for the disc and torus. From these equations, it directly follows that the pizza is a specific 2D linear projection of the torus.
> > >
> > > We believe this question will be of great interest to the interpretability community due to how it can influence discussion about universality. Our findings not only restart discussion about universality, but have the potential to make the discussion: “if DNNs do learn different circuits on the same data, we conjecture the different circuits are always a linear projection of a universal manifold”.
> > >
> > > > 7. Are the authors willing to tone down the claim that "the clock and pizza are the same algorithm", for example, to something like "topological similarities between the clock and the pizza"?
> > >
> > > Yes, we never intended to make this claim, and as stated in weakness 1, we’ve changed our naming conventions to match your suggestions to avoid any future reader confusion. As Fig. 1 shows, the clock and pizza algorithms are totally different, both in topology (clocks have one hole, pizzas have zero), and metric geometry (how points on each manifold are colored and their distances). Rather, our claim is that only one of these algorithms emerges (the pizza). We apologize for the confusion, and hope that our clarifications are sufficient.
> > >
> > > We sincerely thank you for your review. We believe it really helped strengthen the paper. It resulted in two new quantitative tests + PCA variance explained data and improved nomenclature to prevent reader confusion. We look forward to our continued discussion.

---

### Author Response · Authors · 2025-12-03
**Global Rebuttal**

Dear AC,

Thanks for overseeing the review process after recent complications. We want to summarize the state of our work post-rebuttal.

To our knowledge, we're first to study the distribution of phase shifts in the sine functions neurons are known to learn on this task (layer 1 neurons learn $\sin(a + \phi_a) + \sin(b + \phi_b)$ on (a+b) mod n = c. Our ability to correct prior work results from this new view.

Our quantitative analysis used Betti numbers to show Zhong et al.'s claims (trainable attention transformers learn "clocks" in layer 1, while constant (uniform) attention transformers learn "pizzas" in layer 1) were false as nothing with the homology of a "clock" is learned. We also show empirically that neurons have phase shifts distributed uniformly (PADs). Assuming phases are distributed uniformly, we rigorously prove there are two cases. 1. Pizza discs where $\phi_a=\phi_b$ for all neurons (we show empirically that all transformers learn discs). 2. Tori, where $\phi_a \not=\phi_b$ for all neurons.

Since our theorem gives closed-form expressions for the pizza and torus representations, and reviewers asked for further quantitative experiments, we added RSM and CKA (two common methods of comparing manifolds by comparing distances of points on the surface of the manifolds). These two experiments verified empirically that our theorem makes testable predictions and allow us to definitively state that Zhong et al.'s claims are falsified.

**Why is this important?**

- 1. Zhong et al.'s finding that transformers could learn either clocks or pizzas shocked the interpretability community and prematurely ended discussions about the universality hypothesis (that networks trained on similar data will learn similar circuits). Their result was demoralizing as it implied LLMs could learn totally disparate circuits (with no connection between them) in parallel to perform the same task. **This is an issue because if it's true, it means gaining insights from interpretability that generalize or speed up future interpretations is impossible.** They claimed by statistical inference (they did not show it directly) that either clock or pizza circuits are learned in the 1 hidden layer transformers they focused their paper on. **We amend the status quo by showing their claim is false**.

- 2. The universality hypothesis is folklore, it does not have a formal definition. Thus, while intuitively useful, is hard to build on or falsify. We find architectures either learn a torus, or linear projection of a torus (pizza). This allows us to present the first formalized universality hypothesis: *DNNs will learn neural representations corresponding to either a universal manifold, or linear projections of a universal manifold*. **This is important to LLM interpretability because if it's true, it provides a road toward generalizable interpretability.** This follows because once interpretists find the universal manifold DNNs learn for a task, they can find lower rank variants of it by looking for linear projections of it, *i.e* it narrows down the search space.

We believe our work is worth dissemination just from amending the status quo and showing that different transformer architectures do not learn totally disparate and disconnected circuits when trained on the same data. We *strongly* believe our step beyond this, in finding an architecture that **does learn something different (torus)**, and our subsequent mathematical proof that reveals it's "not that different" and that pizzas are just linear projections of tori, is of great interest to the community.

We also think it's beautiful that we connect universality with the manifold hypothesis (Goodfellow et al.). Our work provides a new counter-example to the platonic representation hypothesis as well, since increased depth/width doesn't result in transformers learning the same neural representation as MLP-Concat learns (torus). Thus, these two architectures (with different training objectives, being direct classification (MLPs) and autoregressive next token prediction on sequences (transformers)) are learning representations that embed distances between points differently (falsifies platonic representation). However, the pizza disc is actually the linear projection of the torus that keeps distances between points "the most similar" (this can be seen by the fact that both RSA and CKA both achieve simultaneous high scores for pizza and torus). This provides a view into the nature of platonic representation that goes beyond being a counter-example and shows the intuitions behind it are solid. We can state "if the distances are different between modalities/training objectives, the networks still try to preserve the platonic distances of the universal manifold".

Our work provides the highest resolution interpretation of modular addition performed-to-date, giving a connecting view of three major deep learning hypotheses (universality, manifolds, platonic representation).

Thank you.

---

### Meta-Review · Area_Chair_ysKJ · 2026-01-07

**Summary:**

The work studies the universality of representations hypothesis in the particular setting of modular arithmetic.

Reviews had divergent ratings. However, they agree the questions are important and the approach is interesting. Authors provided an extensive rebuttal which addresses many of the concerns raised in the initial reviews.

In my view the submission contributes a valuable discussion with strengths outweighing weaknesses. Therefore I recommend accept.

**Reviewer Concerns:**

Specify which reviewer concerns you think were addressed by the rebuttal, and which you believe are still outstanding.

Reviewer yR58: concern about the description of the clock and pizza algorithms. The rebuttal offers clarifications, concedes to some of the concerns, and adds experiments.

**Reviewer Scores:**

For each review, specify how you think the reviewer would have changed their score if they had been able to participate fully in the discussion.

Reviewer yR58: 2 -> 4
Reviewer 3muT: 4 -> 4
Reviewer 9zL4: 8 -> 8

---

### Decision · Program_Chairs · 2026-01-26

Accept (Poster)